

# 60 years of UK visibility measurements: impact of meteorology and atmospheric pollutants on visibility

Ajit Singh, William J. Bloss and Francis D. Pope

School of Geography, Earth and Environmental Sciences, University of Birmingham, Birmingham, B15 2TT, UK

*Correspondence to*: Francis D. Pope (f.pope@bham.ac.uk)

**Abstract**

Reduced visibility is an indicator of poor air quality. Moreover, degradation in visibility can be hazardous to human safety; for example, low visibility can lead to road, rail, sea and air accidents. In this paper, we explore the combined influence of atmospheric aerosol particle and gas characteristics, and meteorology, on long-term visibility. We use visibility data from eight meteorological stations, situated in the UK, which have been running since the 1950s. The site locations include urban, rural and marine environments.

Most stations show a long term trend of increasing visibility which is indicative of reductions in air pollution, especially in urban areas. Additionally, the visibility at all sites show a very clear dependence on relative humidity indicating the importance of aerosol hygroscopicity on the ability of aerosol particles to scatter radiation. The dependence of visibility on other meteorological parameters, such as wind speed and wind direction is also investigated. Most stations show long term increases in temperature which can be ascribed to either climate change, land-use changes (e.g. urban heat island effects) or a combination of both; the observed effect is greatest in urban areas. The impact of this temperature change upon local relative humidity is discussed.

To explain the long term visibility trends and their dependence on meteorological conditions, the measured data were fitted to a newly developed light extinction model to generate predictions of historic aerosol and gas scattering and absorbing properties. In general, an excellent fit was achieved between measured and modelled visibility for all 8 sites. The model incorporates parameterizations of aerosol hygroscopicity, particle concentration, particle scattering, and particle and gas absorption. This new model should be applicable and is easily transferrable to other data sets worldwide. Hence, historical visibility data can be used to assess trends in aerosol particle properties. This approach may help constrain global model simulations which attempt to generate aerosol fields for time periods when observational data are scarce or non-existent. Both the measured visibility and the modelled aerosol properties reported in this paper highlight the success of the UK's Clean Air Act, which was passed in 1956, in cleaning the atmosphere of visibility reducing pollutants.

**Keywords:** Visibility, meteorology, pollutants, aerosol, RH and hygroscopicity





## 1 Introduction

The meteorological definition of visibility is the "distance at which the contrast of a given object with respect to its background is just equal to the contrast threshold of an observer" (WMO, 1992, 2003). In general, good visibility is a desirable feature of any geographical location and its importance should not be neglected (Doyle

and Dorling, 2002). Poor visibility can affect the transportation of goods and people, whether it is by rail, road, sea or air. Low visibility can lead to accidents and thus is a concern for public safety. Tourism is often dependent on good visibility for appreciation points of interest (Singh and Dey, 2012). For example a study at Grand Canyon Park in USA has shown that visitor frequency in the park has reduced as visibility decreased (NAPAP, 1990).

Typically in cloud free sky, visibility can vary from ca. 5 - 100 kilometres dependent on atmospheric composition and conditions. Visibility is reduced by the interaction of light with atmospheric gases and aerosol particles which can absorb or scatter the light; consequently visibility is greatest within non-polluted pristine atmospheres, other factors (e.g. meteorology) being equal. Many previous studies have investigated the link between atmospheric composition and visibility (Jinhuan and Liquan, 2000;Schichtel et al., 2001;Wu et al.,

2005;Park et al., 2003;Yang et al., 2007;Tiwari et al., 2011;Park et al., 2006;Founda et al., 2016;Cao et al., 2012;Watson and Chow, 2006). These investigations demonstrate that visibility is markedly influenced by the size, chemical composition, and concentration of airborne particles. Reduced visibility is attributed mainly to high concentrations of aerosol particles, and in general, scattering effects are the dominant visibility reducing mechanism within the atmosphere. Within heavily polluted atmospheres, visibility can decrease rapidly due to

the presence of aerosol particles (Husar et al., 1981). For example, during the 1952 London smog events visibility declined to a few metres due to high air pollution (caused by a rise in smoke and other pollutant concentrations in the atmosphere (Wilkins, 1954)) as discussed in detail by Brimblecombe (1987). More recently, a study by Sati and Mohan (2014) also found sharp changes in visibility due to increased particulate matter (PM) and $NO_2$ concentrations during a smog event on November 2012 at Delhi. Similarly, Zhang et al.

(2006) describe the PM influence upon visibility reduction at Beijing, China. Festivals involving fireworks, which release aerosol particles upon detonation, are a good example of spatially and temporally localised pollution events which may lead to reduced visibility (Singh et al., 2015;Seidel and Birnbaum, 2015;Kong et al., 2015).

In addition to aerosol and gas concentrations and composition, specific meteorological conditions can also affect

visibility (Sloane, 1983). There exists a body of literature on urban visibility studies, which attempt to connect visibility with meteorological parameters (e.g. (Hänel, 1972;Clarke et al., 1978;Lee, 1983;Lee, 1990;Haywood and Boucher, 2000)).Whilst temperature (T), relative humidity (RH), wind speed ($w_s$) and wind direction ($w_d$) do not affect clear sky visibility directly, they can influence the sources and sinks of the trace gases and aerosol particles in the atmosphere. For example, high wind speeds can re-suspend dust particles and generate sea spray

aerosol particles. Windy conditions can also lead to a cleaning effect by replacing polluted air with cleaner air. Temperature can influence the production of secondary organic aerosol (SOA) particles, for example, via the chemical formation and partitioning between the gas and particle phase. Relative humidity (RH) not only affects the sources and sinks of gases and aerosols, it also directly influences the size and composition of aerosol particles. Nearly all atmospheric aerosol particles are hygroscopic to some degree; hence, their size is dependent



upon the local RH. As RH increases, hygroscopic particles take up water, through absorption and adsorption, and grow in size, volume and weight. The addition of water also changes the overall particle composition. This typically lowers the mean refractive index of the particle since the refractive index of water is lower than other common aerosol components, such as minerals, organics, sulphates and nitrates (Harrison et al., 2004). Under high humidity conditions, a high particle loading in the lower atmosphere can increase fog formation and thus severely reduce visibility (Tiwari et al., 2011). It has previously been shown that monthly variations in visibility are negatively correlated with RH (Singh and Dey, 2012). Other studies have shown how the RH effect on particle hygroscopic growth can influence in visibility change (Liu et al., 2012). Thus, both PM loading and meteorological factors, such as relative humidity, are important for the assessment of the causes of visibility reduction. Other factors may also be important such as vegetation density, industrial development, urbanization and human population since these factors affect surface type and can effect aerosol deposition (Diederen et al., 1985).

In the last few years, worldwide interest in atmospheric visibility has grown, but few studies examine UK visibility. Previously, a long-term trend analysis of visibility was performed at eight UK weather stations between 1950 and 1997 by Doyle and Dorling (2002), where improved visibility was identified at most of the stations, mainly after 1973 due to oil crises and less consumption. Summer visibility trends for five different sites in London and southern England for the period of 1962 to 1979 were analysed by Lee (1983), and it was also found that a rise in visibility was observed at all sites. Gomez and Smith (1984) quantified the seasonal visibility trends at Oxford during 1926 to 1985 and observed a clear reduction in visibility from 1926 to 1944, a notable rise after World War II from 1944 to 1952, and another reduction from 1952 to 1966 (mainly in the summer season); the visibility improved again after 1966 in all seasons due to the reduction in aerosol concentration (Gomez and Smith, 1987). It is also found that after the 1956 Clean Air Act, fog occurrence has decline at Oxford and nearby rural areas due to drop in smoke concentration, urban heat Island effect and other public activities (Gomez and Smith, 1984). Analyses by Lee (1985) in central Scotland for the period of 1962 to 1982, has mentioned about the effect of 1973 oil crises on visibility and air quality, where a significant increase in visibility was shown primarily in urban areas due to a major reduction in sulphate aerosol concentration. A similar study on historical visibility trends at 22 different UK meteorological stations (includes urban, rural and marine areas) during 1962 to 1990 was performed by Lee (1994). A clear rise in visibility was identified at most of the sites due to reduction in coal and smoke emissions (Lee, 1994). Furthermore, a steady reduction in fog frequency with improved visibility correlated with decreased smoke pollution at Glasgow airport was noted (Harris and Smith, 1982). The correlation between various air pollutants (such as $NH_4^+$, and non-marine pollutants $SO_4^{2-}$ and $NO_3^-$) and visibility at northwest England, UK were also performed in the 1980s, where strong negative correlations were found between visibility and these pollutants (Colbeck and Harrison, 1984). At present, most UK urban cities are relatively polluted (Defra, 2011) compared to rural locations, with pollutant sources dominated by vehicular emissions (Colvile et al., 2001). The 1956 Clean Air Act led to general improvements in UK air quality; however, there still exist many negative effects of air quality on the UK population such as impaired human health ((Defra, 2011;Harrison et al., 2015).

The present study investigates visibility in the UK focusing on 8 specific sites. The same sites were previously investigated by Doyle and Dorling (2002) who presented long term UK visibility trends for 1950-1997 and the



dependence of the measured visibility on meteorological conditions. In this paper we build upon the work of Doyle and Dorling to analyse UK visibility trends from 1950-2013. Furthermore, we extend the analysis by investigating causes of the observed visibility trends; in particular we investigate the role of air pollutant concentrations upon UK visibility. The outputs from this work help explain historic visibility trends in the UK.

They also provide data with which to generate future visibility predictions based upon UK projections of meteorology and pollutant conditions, and insight into historical atmospheric pollution levels.

## 2 Data

Daily archived horizontal visibility data, defined as the visibility distance along a horizontal line at the earth's

surface, were obtained from the British Atmospheric Data Centre (BADC) which is run by the UK's Natural Environment Research Council (www.badc.nerc.ac.uk). The archive contains visibility data, in addition to other relevant meteorological parameters, archived at an hourly time resolution. In addition to visibility, the following meteorological parameters were also utilised: RH, wind speed, wind direction and air temperature, present weather (PR) code which provides further qualitative detail about the weather conditions. A description of the

present weather codes is provided in the table (www.badc.nerc.ac.uk/data/ukmo-midas/WH_Table.html) at www.badc.nerc.ac.uk. Unfortunately the use of present weather codes largely ceased with the introduction of automated meteorological stations and insufficient PR codes were available after the year 1997. It is noted, that if the present weather codes were available they would have been useful to screen the data for rain or other precipitation events. Due to unavailability of present weather codes during required study period (1950-2012),

data filtration was done on the bases of RH limits instead of PR codes. Data were removed when the relative humidity reading was > 99 % which is highly suggestive of rain or other precipitation events. Since the ability of visibility observers is affected by light levels, with greater difficulty encountered in night time measurements (Lee, 1990) the daily data used for this study were all measured at 12 noon for all sites.

Meteorological data were collected for the eight UK stations which possess near continuous time series data

starting in the 1950s and continuing to present day. The eight stations are Aldergrove, Heathrow, Ringway, Nottingham, Plymouth, Tiree, Leuchars, and Waddington, and details of the stations are given in Table 1 and Fig. 1.

The visibility data sets are based on ground based measurement using a variety of techniques. Until the late 1990s all visibility measurements were performed by human observation. Subsequently data collection was

automated using visibility sensors (visiometers). See supplementary material Table S1 for detail on measurement type used and dates of service.

There are advantages and disadvantages with both manual observation and visiometers. Clearly from a manpower perspective, visiometers are preferred. Manual observation provides a true measure of visibility since the observer is looking for objects located at a known distances away from their location, however, the visibility

measurements are imprecise by nature since results can vary according to the contrast and illuminance





thresholds (ability to discern and sensitivity to light, respectively) of the observer's eyes (WMO, 2008). Since manual observation requires objects to observe the measurement is quantized by the geographical spread of available objects i.e. there is not a continuum of measurement locations. Consequently, human observations provide a lower limit to the actual visibility. Distances between objects to observe can be large especially at the

5 longer distances measured (> 10 km) which leads to reductions in accuracy at high visibility. At high elevation the visibility calculation can be different from that at the surface (Malm et al., 1981). Visiometers automatically measure the extinction of light over a small distance (typically ca. 1 m) and from the measured extinction can estimate visibility. In particular automatic visibility measuring instruments consist of a light transmitter and receiver, the light extinction observed between the transmitter and receiver is then used to estimate the visibility

(Jebson, 2008). These automated estimates of visibility are more objective and reproducible compared to than human observation. However, since the visiometer only measures air local to the device it can be much more affected by variations in local air quality. This is likely to be a more important consideration at urban meteorological sites where air composition is more heterogeneous, compared to rural sites, due to the greater number of pollutant sources in urban areas.

The change from manual to automatic measurement occurred at different times for the different sites (see supplementary Table S1). It is clear for most sites investigated, that the changeover from manually observed to automatically measured data leads to step changes in the visibility reported, see Fig. 2 and further discussion in methodology section. This is unsurprising given the discussion above. In particular, clear deviations away from the long term trend measured under manual observation are observed at Aldergrove, Plymouth and Tiree

stations once automation was introduced (see supplementary Table S1). After consultation with the UK Met Office it was noted that automated sensors can be unreliable during high visibility events when compared to manual readings. In particular automatic sensors perform sub-optimally at coastal sites unless the sensor is cleaned regularly, due to accumulation of sea salt residue. Unfortunately, the Tiree station was reported to fall into this category.

To assess effects of the gaseous pollutant nitrogen dioxide ($NO_2$) on visibility, daily ground based measured data of $NO_2$ was obtained from the Department of Environment Food and Rural Affairs (Defra) (https://uk-air.defra.gov.uk/) for one observing station (Harlington), closely co-located to the Heathrow meteorological station (ca. 1.3 miles distant). $NO_2$ data were only available for 9 years (2004-2012) of the visibility study period.

## 3 Methodology

### 3.1 Trend analysis of visibility and other meteorological parameters

60 year trend analyses have been performed on the visibility dataset described in section 2. For long term trend analysis each days value was averaged (mean) to determine trends over decadal, annual and seasonal cycles.

The seasonal periods were defined, as is typical, as winter (Dec-Feb), spring (Mar-May), summer (Jun-Aug), and autumn (Sep-Nov). Diurnal, day of the week and monthly averaged trends of visibility and RH were





determined at each site using the 60 years of dataset, where weekdays and weekend are categorised as Monday-Friday and Saturday-Sunday respectively.

To examine the hygroscopic growth effect of aerosol particles upon visibility, the decadal data sets were disaggregated into RH bins. The aerosol hygroscopic growth effect on visibility was examined by using decadal mean visibility within specific relative humidity bins with the following boundaries: 52.5-57.5 %, 57.5-62.5 %, 62.5-67.5 %, 67.5-72.5 %, 72.5-77.5 %, 77.5-82.5 %, 82.5-87.5 %, 87.5-92.5 %, 92.5-97.5 %. We excluded data with RH > 97.5 % due to likely presence of fog and mist at RH greater than this threshold.

To highlight the daily variation in RH, histograms of daily RH (at 12 noon) were generated using the following boundaries (0-10 %, 10-20 %, 20-30 %, 30-40 %, 40-50 %, 50-60 %, 60-70 %, 70-80 %, 80-90 %, and 90-100 %).

To evaluate the dominant meteorology at each site several meteorological analyses were conducted. Wind rose plots using the complete dataset time series were generated to highlight the dominant wind speed and direction for all sites. Decadal-seasonal bivariate polar plots of visibility using wind direction and wind speed allow for spatial analysis of likely pollution sources (Carslaw and Ropkins, 2012). Finally time series plots of the following meteorological parameters were generated, RH, wind speed, wind direction and air temperature.

### 3.2 Estimation of aerosol and gas phase properties through analysis of RH dependent visibility

In this section the contribution of aerosol particles and gases upon visibility is estimated via mathematical modelling. In general horizontal visibility ($V$) can be defined via Koschmieder equation Eq. (1), where, horizontal visibility shows an inverse relationship with the extinction coefficient ($\beta_{ext}$). In the Eq. 1, constant (k) is equal to 3.912 which assumes a contrast threshold of 2 % (Koschmieder, 1924). The constant (k) is a measured by the threshold sensitivity of the observer's eye (Schichtel et al., 2001;Chang et al., 2009), which can vary from 2 to 5 % (Appel et al., 1985).

$$V = \text{k}/\beta_{ext} \tag{1}$$

The extinction coefficient depends upon ($\beta_{ext}$) is the sum of the scattering ($\beta_{sca}$) and absorption coefficients ($\beta_{abs}$) as shown in Eq. (2).

$$\beta_{ext} = \beta_{sca} + \beta_{abs} \tag{2}$$





In the atmosphere, aerosol particles and gas phase species can both contribute to light scattering and absorption. However, the contribution of gas phase scattering to the total extinction is negligible except in the most pristine environments. Hence under UK conditions, the scattering component of the extinction coefficient can be assumed to be completely dominated by the presence of aerosol particles.

5    The ability of an individual particle to scatter radiation is dependent on its size, shape, morphology and refractive index (Appel et al., 1985;Liu and Daum, 2000). The particle scattering coefficient ($\beta_{sca}$) can be estimated by Mie theory as shown in Eq. (3) (Tang, 1996);

$$\beta_{scat} = \int_0^\infty \pi \left(\frac{D}{2}\right)^2 Q_{scat}(\alpha, \lambda, n) N f(D) dD \qquad (3)$$

Where, $D$ represents particle diameter, the aerosol size distribution is given by N$f(D)$ and α is the size parameter (α = πD/λ). N is particle number concentration and $Q_{scat}(\alpha, \lambda, n)$ is single-particle scattering cross section, which depends upon size parameter ($\alpha$), wavelength (λ) and refractive index ($n$, which is composition dependent). All these particle characteristics can change as the particle undergoes water uptake or loss which is

15    dependent on the local RH. To parameterise the aerosol scattering enhancement due to water uptake an approach, similar to Titos et al. (2014), is taken. The scattering enhancement is parameterised using a single hygroscopicity parameter ($\gamma$) using Eq. (4), where $\beta_{sca}(RH)$ and $\beta_{sca}(dry)$ are the aerosol scattering coefficients under a specified RH condition and completely dry conditions, respectively.

$$\frac{\beta sca(RH)}{\beta sca(dry)} = \left(1 - \frac{RH}{100}\right)^{-\gamma} \qquad (4)$$

Rearranging Eq. (1), Eq. (2), and Eq. (4) allows for the relationship in Eq. (5) to be derived, where $\beta_{abs}(RH)$ and $\beta_{abs}(dry)$ are the combined aerosol and gas absorption coefficients under a specified RH condition and completely dry conditions, respectively.

$$Vis(RH) = \frac{3.912}{\left(1 - \frac{RH}{100}\right)^{-\gamma} \times \left(\frac{3.912}{Vis(dry)} - \beta_{abs(dry)}\right) + \beta_{abs(RH)}} \qquad (5)$$

To reduce the number of parameters within Eq. (5), it is assumed that $\beta_{abs}(RH) = \beta_{abs}(dry)$. This assumption always holds for gas absorption; and it is largely true for aerosol particles as well, although it is noted that

30    particle absorption can increase due to lensing effects in mixed phase aerosol, and this lensing effect will be affected by aerosol water content e.g. (Lack and Cappa, 2010).



Equation (5) can be further simplified by assuming that all absorption due to both gases and particles is negligible compared to the RH dependent aerosol scattering, leading to the two parameter Eq.(6).

$$\log [Vis(RH)] = \gamma \log \left[ 1 - \left( \frac{RH}{100} \right) \right] + \log [Vis(dry)] \qquad (6)$$

Equations (5) and (6) can be used to obtain information about aerosol scattering and gas and aerosol absorption, with associated assumptions, through fitting of the measured visibility at a given RH. Equation (6) is linear and so can be fitted using the linear least squares fitting algorithm, whereas Eq. (5) requires non-linear least squares fitting algorithm. The statistical program R was used for all fittings (Version 0.99.489). The 'lm' algorithm was 10 used for linear fitting, and the 'nls' fitting algorithm was used for the non-linear fitting. The 'nls' algorithm was always initially run with no lower or upper boundaries for the 3 fitting parameters ($Vis(dry)$, $\beta_{abs}$ and $\gamma$) specified. However, when fits produced negative values for $\beta_{abs}$, which are physically impossible, a lower boundary for $\beta_{abs}$ was specified to be zero.

15  **3.3 Gas absorption**

All gases scatter radiation via Rayleigh scattering but the effect is negligible in all but the most pristine visibility conditions (which are not observed in this study). The only atmospheric gas present at levels that lead to significant absorption of visible light is $NO_2$ (Ferman et al., 1981;Groblicki et al., 1981). The contribution of $NO_2$ to visibility can be quantified by its absorption coefficient ($\beta_{NO_2 abs}$). The effect of the $NO_2$ absorption 20 coefficient, at 550 nm wavelength, was calculated using the relationship from Groblicki et al. (1981), shown in Eq. (7), where $[NO_2]$ is the $NO_2$ in ppm.

$$\beta_{NO_2 abs} = 3.3 \times 10^{-4} [NO_2] \qquad (7)$$

**4. Results and Discussion**

25  **4.1 Historical trend of annual and seasonal visibility**

The annual and seasonal mean visibility at 12 noon have been calculated for all eight stations, see Fig. 2. The effect of changing the visibility observation technique from manual observation to automatic observation via visiometers (which is highlighted by different shading in Fig. 2) is very clear at some sites. In particular, two stations, Tiree and Aldergrove, do not show realistic values after the changeover from manual to automated 30 measurement, with the changeovers coinciding with large and sustained drops in recorded visibility. The effect of manual to automated changeovers at Heathrow, Leuchars, Nottingham, Ringway and Waddington sites





appears to be minimal, with the pre-changeover long term trends being continued after the changeover. Furthermore the annual data from these sites exhibit similar year to year variance before and after changeover. The long term trend at the Plymouth site is similar before and after changeover but the year to year variance is much reduced once measurement automation is installed. This likely indicates strong localised sources close to the visiometer at the Plymouth site. Henceforth it is assumed that all stations, except Aldergrove and Tiree, are performing adequately for both manual and automated visibility measurement. Therefore the time series, as shown in Fig. 2, are used in their entirety for the analysis of these six stations. The time series data for the Aldergrove and Tiree stations are used up until automation occurs.

Clear trends of increasing annual visibility are observed for four sites: Ringway, Waddington, Nottingham, and Heathrow with the rate of visibility increase being $0.339\pm0.016$ km year$^{-1}$, $0.293\pm0.010$ km year$^{-1}$, $0.235\pm0.023$ km year$^{-1}$ and $0.201\pm0.018$ km year$^{-1}$, respectively, where standard errors were determined at the 95 % confidence interval. A more gradual increasing trend was observed at the Leuchars site ($0.157\pm0.019$ km year$^{-1}$). The Plymouth site shows a more variable trend with increases from ca. 1950-1990 followed by decreases from ca. 1990-2006 which is then followed by more increases in the most recent measurements. The long term trend for Plymouth 1950-2013 is near constant ($0.040\pm0.021$ km year$^{-1}$). Both the Aldergrove and Tiree sites, with the automated data omitted, show near constant long term visibility with long term rates of visibility change calculated to be $0.0562\pm0.021$ km year$^{-1}$ and $-0.0892\pm0.014$ km year$^{-1}$, respectively.

The seasonal trends for the 8 sites are detailed in Table 2. Poorest visibility was observed in the winter season compared to other seasons mostly due to the seasonal rise in RH (discussed in section 4.3). Another reason is the greater concentration of particles in the environment due to lower mixing layer height in the winter season (Jayamurugan et al., 2013). Furthermore, the long term rate of visibility change in the winter season is significantly higher as compared to spring, summer and autumn seasons for all stations apart from the Ringway station. At Ringway station the rate of change of visibility is higher in spring ($0.363\pm0.018$ km year$^{-1}$) as compared to winter ($0.330\pm0.020$ km year$^{-1}$). All stations show positive rates of visibility change in winter season except for Tiree ($-0.186\pm0.012$ km year$^{-1}$). It is also observed that Aldergrove station shows negative rate of visibility change in the summer season ($-0.417\pm0.036$ km year$^{-1}$).

### 4.2 Evaluation of historical wind-data

### 4.2.1 Wind Roses for the 8 stations

A graphical representation of historical wind speed and direction at the eight chosen stations is shown in Fig. 1 using the wind rose polar co-ordinate representation. These graphs describe the most probable wind speeds and directions over the whole time series (Carslaw and Ropkins, 2012). As expected, the graphs show that the predominant wind directions in the UK are from the southwest. However, there are clear variations between the different stations. The range of wind speed varies from 0-35 m s$^{-1}$ dependent upon location, with the more coastal sites experiencing greater average wind speeds.





### 4.2.2 Analysis of influence of wind speed and wind direction on visibility

Decadal-seasonal bivariate polar plots are presented for all eight stations in supplementary Fig. S2; these diagrams provide information on the variation of visibility with wind speed and direction and can suggest locations for visibility degrading sources. The detailed analyses of each site are given below:

**Aldergrove**: Overall, lower values of visibility were observed when the wind was from the south to east, while above average values were collected when the wind was from the north to west direction. Intermediate visibility was generally observed when the wind came from the south to west or north to east quadrants. Distinct differences are observed between the different seasons. In particular, in the summer visibility with wind from the north to west direction was higher compared to other seasons in every decade. It is clearly seen that visibility

has improved the most when wind comes from the south to east direction which covers mainland urban areas - in particular, higher wind speeds from the direction of Belfast, the major regional city, leads to lower visibility over Aldergrove. It is noted that the seasonal and polar trends are similar between the visiometer (1950s–1990s) and human derived (2000s–2010s) data sets even though the absolute magnitudes are different as noted above.

**Heathrow:** Low visibility was observed whenever wind speeds were lower than 5 m s$^{-1}$ in any direction which

implies a significant local source of visibility degrading pollutants. Since Heathrow is the site of major international airport, with commensurate road and other transport infrastructure, this is not surprising. Overall, lower visibility is also seen when the wind direction comes from the northeast to southeast direction which is consistent with visibility reducing pollution arriving from the Greater London area. The highest visibilities are typically observed when the wind direction is from the north to southwest which is consistent with less densely

populated surrounding areas. In particular during summer visibility in the northwest wind direction was highest compared to other seasons in every decade. It is identified that visibility has improved in all wind directions, but most significantly in the easterly direction which covers the London urban centre. The change in visibility illustrates the dramatic improvement of air quality in London since the introduction of the Clean Air Act in 1950s (Brimblecombe, 2006).

**Leuchars:** Two distinct spatial groupings of visibility are clearly observed. When the wind direction comes from the northeast to southwest (clockwise) visibility is generally lower, and it is generally higher when the wind direction is from the northeast to southwest (anti-clockwise). The lowest visibilities are from the southeast direction in all seasons. The spatial pattern of low visibility suggests a maritime aerosol source as the major source of visibility reduction whilst high visibility was associated with air which had passed over the

predominantly rural Scotland. Visibility in the northwesterly wind direction was highest in the summer months, as expected see Fig. 2 and 3, compared to other seasons in every decade.

**Nottingham:** Like Heathrow, the poorest visibility conditions occurred when wind speed was below 10 m s$^{-1}$ suggesting local sources of visibility degrading pollutants. Visibility is often lowest when the wind comes from the southeast direction consistent with the relative placement of Nottingham city centre to this direction (the

35 meteorological station is actually located in Watnall just about 5 miles of Nottingham city centre). Visibility is generally highest when the wind comes from the west and southwest directions which is largely consistent with air masses passing over less urban areas compared to the other wind directions. During the summer months,





visibility in southwest direction was highest compared to other seasons in every decade. It is clear from Fig. 2 that visibility has increased in all seasons, and the strongest improvement is seen in air from the southeast as seen in supplementary Fig. S2.

**Plymouth:** In general, the lowest visibility was observed when the wind comes from southeast to southwest direction which is consistent with maritime air causing the lowest visibility which suggests a maritime source of aerosol causing visibility degradation. The highest visibilities are observed when wind comes from the northwest to northeast directions, and in particular the northeast, this is consistent with airmasses passing over relatively rural areas. Regardless of the direction of wind, the summer months showed higher visibility than all other seasons. It is identified that visibility has improved over time for all wind directions.

**Ringway:** Overall visibility was poor at low wind speeds and when the wind direction was from the northeast to southeast. Ringway is the location of Manchester International Airport so, like Heathrow, there is likely to be a significant local source of visibility degrading pollutants arising from the airport and its associated infrastructure. The wind directions associated with higher visibility are a lot more variable in time and space when compared to other locations. However, in general, high wind speeds from either the northwest or south west directions are often associated with higher visibility. Since the 1960s visibility has improved for all wind directions. In particular, visibility associated with air masses coming from the direction of the Greater Manchester Area to the north has shown a marked increase since the 1970s.

**Tiree:** The island of Tiree has by far the highest visibility at low wind speeds. Overall low visibility was observed when wind came from the west to southeast, while highest visibility occurred with wind from the northeast. The spatial variation of low visibility is consistent with a maritime source of visibility impairing aerosols. The higher the wind speed typically the lower the visibility which is consistent with greater aerosol production from greater wave activity (Venkataraman et al., 2002). The higher visibility from the northeast is consistent with air masses passing over the larger rural highlands of Scotland. Visibility was relatively stable for all wind directions for all decades of the manual observation data series which is consistent with this rural maritime site being largely unperturbed by anthropogenic pollution.

**Waddington:** In general, lower visibility is observed when wind speeds are lower than 10 m s$^{-1}$ which is consistent with local pollution sources. Low visibility is also observed when the wind direction is from the east to southeast which potentially indicates a maritime source. Higher visibility is observed from the west at high wind speeds. Visibility has improved for wind from all directions since the 1970s.

Overall it is clear that visibility has improved at most of the sites for most local wind directions. The most marked improvements in visibility are seen in directions when air masses pass over major metropolitan areas such as Greater London and Greater Manchester. Whilst most of the visibility changes can be ascribed to the location of the meteorological stations with respect to either urban or maritime sources, it is noted that for most sites the wind direction with the lowest visibility overall is often from the East, i.e. continental Europe and hence synoptic scale pollution events which affect visibility. Poor air quality, in the UK, is often associated with synoptic scale events originating in continental Europe (Charron et al., 2007a;Charron et al., 2007b;Charron et al., 2013;Lee et al., 2006)





### 4.3 Correlation between RH and visibility: seasonal, day of the week and decadal effects

Figure 3 provides monthly values for visibility and RH, averaged over the whole time series, for each station. This figure clearly illustrates that visibility shows a strong seasonal cycle which is anti-correlated with RH at all stations. The relationship at Tiree is less strong compared to the other seven sites. The geographical location of Tiree, which is a maritime island, is the likely reason for the RH trend being different to the other stations. Tiree Island has a very flat landscape, which does not provide shelter from wind in any direction; this directly affects the local meteorology (Holliday, 2004). Overall, the monthly trends indicate that visibility is lowest in winter and highest in summer with spring and autumn being intermediate in visibility values.

In addition to the seasonal cycle, there is a clear day of the week effect on visibility changes at most sites (Fig. 3), where visibility improves sharply at the weekend with Sunday showing the highest visibility. It is observed that visibility improves at Sunday from 5 % to 12.5 % (depending upon area) as compared to other week days (Mon-Fri). Lower traffic and industrial emissions at the weekend are the likely reasons for better visibility due to less pollutant emissions. The visibility peak on Sunday (rather than Saturday and Sunday) may reflect the timescale for pollutant removal by wind driven dispersion. A typical wind speed of 10 m s$^{-1}$ (see Fig. 5) is equivalent to 86 km day$^{-1}$, hence visible air is typically dispersed by wind on a timescale of less than 1 day. The same argument explains why visibility is typically higher on Mondays compared to the other working week days. The inherent assumption in this analysis is that traffic is higher during week days compared to the weekend.

The long term decadal (1950s–2010s) variation in visibility with RH is shown in Fig. 4, for all 8 stations, where the visibility is averaged within RH bins. A qualitatively similar pattern has been observed for all stations: Visibility is observed to vary strongly with relative humidity, which clearly indicates a significant particle hygroscopicity effect on visibility. It is noted that very high RH can also be indicative of precipitation which also decreases visibility.

To further highlight the effect of RH on visibility, the mean monthly visibility trend is compared to RH for the 60 years of data recorded at the Waddington station, see supplementary Fig. S1. A scatter plot of visibility versus RH reveals a clear near-linear relationship (linear fit $R^2 = 0.60$) between the variables. Removal of the long term trend in the visibility data was achieved by fitting the visibility to a quadratic function and subtracting the quadratic function from the time series. A scatter plot of the long term detrended visibility data versus RH reveals a more linear relationship ($R^2 = 0.66$) where every rise in RH of 10 % results in a reduction of approximately 5 km of visibility.

### 4.4 Effect of long term changes in meteorological parameters upon visibility

The long term trends in visibility are compared to the other recorded meteorological parameters: RH, air temperature, wind speed and wind direction (Fig. 5). It is observed that at most of the stations RH decreases as average air temperature increases. Previous literature observed that the UK mean air temperature and sea surface temperature have increased by about 1$^\mathrm{o}$C and 0.7$^\mathrm{o}$C respectively between the early 1970s and mid 2000s





(Jenkins, 2007). However, overall UK mean RH decreased about 2.7 % between the 1961 and 2006 (Jenkins, 2007). This reduction in RH is also seen more widely in the mid-latitudes (Willett et al., 2014). The temperature change is likely due to climate change, land-use (urban heat island) effects or a combination of both. Clearly, urban heat island effects can only affect stations that are located in urban areas (Fig. 5). However, as Fig. 5

shows, visibility is strongly related to relative humidity and hence to the air temperature of a given location, highlighting a possible indirect effect of climate change and urban heat island effects on regional visibility.

**4.5 Mathematical fitting of measured visibility**

Equations 5 and 6 were fit to the decadal visibility data subset into distinct RH bins, as detailed in section 3.2. It

is found that the fitted data is able to match the observed visibility extremely well ($R^2 > 0.98$) for all stations; for example see Fig. 6 for Heathrow station. The last decade, starting in 2010, has the poorest fit, albeit still with an $R^2 = 0.95$, but only comprises 3 years of data.

We have quantified, in section 4.1, that the decadal observed visibility has improved at most of the stations, which is a direct indicator of change in the combination of aerosol concentration, aerosol composition, gas

concentration and RH. To better understand these changes in visibility, the absorption coefficient ($\beta_{abs}$), scattering coefficient ($\beta_{sca}$), particle hygroscopicity parameter ($\gamma$), and dry visibility ($Vis(dry)$) have all been calculated via constructed model (Eq. (4)) described in section 3.2.

The determined model output parameters ($Vis(dry)$, $\gamma$, $\beta_{sca}$, and $\beta_{abs}$) are presented in Fig. 7, where analysis has been carried out for all sites within each decade; however, the following discussion only considers data that was

measured manually, due to the impacts of measurement methodology changes noted above. A clear improvement in calculated dry visibility was observed for Plymouth, Heathrow, Ringway, Nottingham and Waddington, while only minor changes were observed at Aldergrove, Leuchars and Tiree (Fig. 7a and supplementary Table S2). Broadly, the 5 sites in England are similar with all showing an upwards trend in visibility, whereas the Scottish and Northern Irish sites have greater dry visibilities but less discernible trend

with time.

The derived value for $\gamma$ has decreased slightly at Heathrow, Leuchars and Ringway sites over those decades (Fig. 7b and supplementary Table 2), which indicates a decrease in hygroscopicity over the time (and a concomitant improvement in visibility). Tiree is the only station which showed increased hygroscopicity parameter values, implying a rise in aerosol particle hygroscopicity which results in a drop in visibility. The

other stations like Aldergrove, Ringway, Plymouth, and Waddington show very little change in hygroscopicity parameter values.

Reductions in scattering coefficient are observed at all sites except Aldergrove. The scattering coefficients calculated at RH = 75 % is shown in Fig. 7d. Larger decreases in the scattering coefficient are observed at the

urban sites compared to the rural sites. Reductions are also observed in the absorption coefficient at most sites but there is much more variability compared to the scattering coefficient. It is interesting to note that the two



most remote sites, both in Scotland, have increasing absorption coefficients, which is potentially indicative of episodes of long range transport of absorbing aerosol to these pristine sites becoming more frequent. As expected, both the absorption and scattering coefficients show an inverse relationship with the observed visibility (Fig. 7a and 7c).

The change in the fitted values for dry visibility and scattering coefficient are not significantly affected by the change in visibility measurement from manual observation to visiometers. Contrastingly, the absorption coefficient and gamma values are much more influenced by measurement technique. This likely indicates that local sources have markedly different absorption and hygroscopicity parameters compared to more regional sources; whereas their local and regional scattering properties are relatively similar.

The modelled scattering coefficient, at 75 % RH, is always higher than the absorption coefficient for all sites and times. However, at lower RH the two values become more comparable, see supplementary Fig. S3 which examines the contribution of the scattering coefficient to the total extinction coefficient at Heathrow. The non-negligible contribution of the absorption coefficient to the total extinction coefficient indicates that the model shown in Eq. (5) is not appropriate for the data reported in this paper. However, for other locations with lower
concentrations of absorbing species, gas or aerosol, the model may be valid and the benefit of a linear fitting algorithm, compared to a non-linear algorithm, could be exploited. It is shown the contribution of aerosol scattering to total extinction has remained relatively constant over time which indicates that the reduction in particulate matter has decreased both the absorbing and scattering fractions in equal measure.

Seasonal decadal changes in aerosol parameters were calculated for the Heathrow station (supplementary Fig.
S4). In general, an improved dry visibility with reduced $\beta_{abs}$ and $\gamma$ values was observed for all seasons over time. However, during winter months the greatest improvement in dry visibility with a reduction in $\beta_{abs}$ was noted.

Trends in visibility for those data acquired at a single RH value of 70 % (67.5-72.5 %) during the period of 1950s to 1990s were investigated for the Heathrow site to demonstrate the disaggregation of the RH effect on
visibility from aerosol concentration effect upon visibility. At constant RH, a clear improved visibility was determined for the study period (supplementary Fig. S5). The result implies significant changes in aerosol composition/concentration are driving the visibility trend. Hence improving air quality contributes significantly to better visibility.

**4.6 Effect of nitrogen dioxide gas upon visibility at Heathrow**

The potential influence of $NO_2$ levels upon visibility was analysed using data from the Harlington station (proximate to the Heathrow site), for the period 2004 - 2012. The annual mean concentration of $NO_2$ varied from 33.6 μg m$^{-3}$ to 38.5 μg m$^{-3}$, peaking in 2005 (Table 3). The $NO_2$ influence on observed visibility (in the RH bin centred at 75 % (72.5-77.5 %)) was greatest in the year of 2005 (where it contributed 4.7 ± 1.6 % in total
extinction) and lowest for 2012 (3.3 ± 1.5 % in total extinction) with the remaining visibility reduction being caused by aerosol extinction. Overall, during 2004 to 2012 $NO_2$ contributed approximately 4 % to the observed visibility change, while the remaining 96 % contributed arose from aerosol particles and fog. However it is





worth considering the contribution of $NO_2$ towards the total extinction coefficient during the 1970s when visibility was very low (16.5 km) as compared to 2012 (25.24 km) and $NO_2$ levels higher. Unfortunately $NO_2$ data is not available before 2004 at nearby Heathrow site, but a recent study shows that, $NO_x$ emission in UK has almost doubled in the time period 1970 to 2012 (Harrison et al., 2015). Using the UK $NO_x$ record for 1970

from Harrison et al. (2015), we assumed the annual mean $NO_2$ concentration in 1970 is double what is measured in the year 2012 (34.6 µg m$^{-3}$) as emission estimates are approximately related to concentration. This assumption does not take into account the changing vehicle fleet with corresponding changing emissions of NO and $NO_2$ (Carslaw and Rhys-Tyler, 2013). Using this data the absorption coefficient for $NO_2$ was calculated. In particular, a higher absorption coefficient ($\beta_{NO2abs}$) in 1970 (0.0121 km$^{-1}$) as compared to 2012 (0.00507 km$^{-1}$)

was identified. However, the contribution of $NO_2$ to the total extinction coefficient remained at 5.2 % in 1970, only about 2 % higher than in 2012.

### 4.7 Conclusions

Long term trends in visibility for 8 meteorological stations situated in the UK have been investigated. In general,

visibility has improved at most of the stations through time. The improvements are greatest in urban areas, and are attributed to reductions in aerosol particle loadings and decreases in atmospheric RH. Visibility was found to be lowest during winter and highest in the summer due to seasonal variations in RH and likely changes in the mixing layer height. The rate of change of visibility was higher in winter for all stations, with the exception of Ringway. A sharp positive increment (5-12.5 %) in visibility was observed on Sundays, as compared to other

20 days of the week (Mon-Sat), which is most likely due to weekend reductions in traffic and other particulate matter emission sources.

Bivariate polar plots of visibility, which account for both the influence of wind speed and wind direction, explained the influence of wind on likely source areas of visibility reducing aerosols. These bivariate polar plots identified likely locations for visibility reducing pollutants sources and their variation over time. Overall, an

25 improved visibility at most of the stations in almost all directions was observed with notable improvements when the air masses moved over metropolitan areas, for example, Greater Manchester and Greater London Areas. At most sites, low visibility was observed when the winds came from the direction of continental Europe which may indicate an influence of regional pollution events leading to visibility reductions. Significant changes in visibility were observed with changes in relative humidity, which indicates a strong dependency of visibility

on aerosol hygroscopicity. The measured RH at all sites was typically in the range of 60-80% and variations of a few percent in this RH range can have significant effects on visibility. Many sites showed long term decreases in RH which correlated with increases in air temperature, and had the effect of improving visibility. If the trend of increasing RH continues, the UK can expect further improvements in visibility for the same pollutant loading.

Calculations indicate that the majority of visibility reduction is caused by PM, however, a non-negligible

contribution of light absorption is due to $NO_2$ gas. For the Heathrow station, over the time period 2004-2012, light absorption by $NO_2$ was calculated to contribute approximately 4% to the total visibility reduction, with the





remainder caused by PM absorption and scattering. The $NO_2$ contribution was likely to have been significantly higher in prior decades due to the higher $NO_x$ emissions and hence atmospheric concentrations.

A light extinction model was developed to explain the dependency of visibility upon meteorology and aerosol characteristics. The agreement between the modelled and measured visibility is excellent. The model suggests that there have been significant changes in aerosol concentration over the last 60 years. The model incorporates parameterizations of aerosol hygroscopicity, particle concentration, particle scattering, and particle and gas absorption. The developed model is easily transferrable and applicable to other data sets worldwide.

Visibility can be used as a proxy for aspects of air quality, in particular particulate matter and nitrogen dioxide. Since visibility measurements can extend back for hundreds of years whilst air quality measurements typically only go back decades albeit with a few sparse datasets going back longer in time. The approach demonstrated in this paper has potential for generating historical air quality indications for locations with visibility records.

*Acknowledgments*

We thank the University of Birmingham for supporting Ajit Singh through the Elite Scholarship Scheme. We are also thankful to UK Met Office and DEFRA for the provision of data used in this research.





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





**Tables**

**Table 1** Study stations with area and length of data description

| No. | Station Name | Station code (src id) | Area | Period | Length of Data (in Year) |
|---|---|---|---|---|---|
| 1 | Aldergrove | 1450 | Urban (Airport) | 1950-2012 | 63 |
| 2 | Heathrow | 708 | Urban (Airport) | 1950-2012 | 63 |
| 3 | Ringway | 1135 | Urban (Airport) | 1950-2004 | 55 |
| 4 | Nottingham | 556 | Urban | 1957-2012 | 56 |
| 5 | Plymouth | 1336 | Urban (near coastal area) | 1950-2012 | 63 |
| 6 | Tiree | 18974 | Rural (Airport, near Coastal area) | 1957-2012 | 56 |
| 7 | Leuchars | 235 | Rural (RAF, near coastal area) | 1957-2012 | 56 |
| 8 | Waddington | 384 | Rural (RAF, Airport) | 1950-2012 | 63 |

\* RAF stands for Royal Air Force

**Table 2** Rate of change of visibility (in kmyear$^{-1}$) with their standered error at 95% confidence interval

| Satiation | Year | Annual | Winter | Spring | Summer | Autumn |
|---|---|---|---|---|---|---|
| Plymouth | 1950-2012 | 0.040 ± 0.021 | 0.152 ± 0.017 | 0.006 ± 0.025 | -0.043 ±0.031 | 0.049 ± 0.022 |
| Aldergrove | 1950-2002 | 0.056 ± 0.021 | 0.110 ± 0.019 | 0.831 ± 0.030 | -0.417 ±0.036 | 0.074 ± 0.029 |
| Heathrow | 1950-2011 | 0.201 ± 0.018 | 0.231 ± 0.021 | 0.181 ± 0.020 | 0.145 ± 0.028 | 0.226 ± 0.020 |
| Ringway | 1950-2004 | 0.339 ± 0.016 | 0.331 ± 0.020 | 0.363 ± 0.018 | 0.316 ± 0.025 | 0.343 ± 0.018 |
| Waddington | 1950-2012 | 0.293 ± 0.010 | 0.331 ± 0.019 | 0.245 ± 0.016 | 0.270 ± 0.018 | 0.325 ± 0.016 |
| Leuchars | 1957-2012 | 0.157 ± 0.019 | 0.286 ± 0.027 | 0.140 ± 0.030 | 0.030 ± 0.034 | 0.180 ± 0.025 |
| Tiree | 1957-2002 | -0.089 ± 0.014 | -0.186 ± 0.014 | -0.035 ± 0.015 | -0.098 ±0.015 | -0.046 ± 0.015 |
| Nottingham | 1957-2012 | 0.235 ± 0.023 | 0.293 ± 0.022 | 0.214 ± 0.024 | 0.149 ± 0.033 | 0.270 ± 0.022 |





**Table 3** Gases contribution in visibility change over Heathrow airport

| Year | NO$_2$ concentration (µg m$^{-3}$) | NO$_2$ ( ppm) | Total Extinction coefficient (km$^{-1}$) by all effects (using E1) | Absobption coefficient in km$^{-1}$ ( $\beta_{NO2abs}$ ) by NO$_2$ | % contribution of NO$_2$ in total extinction cofficient |
|---|---|---|---|---|---|
| **2004** | 38.3 | 0.0203 | 0.1475 | 0.00671 ± 0.0023 | 4.5 ± 1.5 |
| **2005** | 38.5 | 0.0204 | 0.1425 | 0.00675 ± 0.0023 | 4.7 ± 1.6 |
| **2006** | 36.9 | 0.0196 | 0.1978 | 0.00648 ± 0.0022 | 3.3 ± 1.1 |
| **2007** | 36.9 | 0.0197 | 0.1855 | 0.00649 ± 0.0029 | 3.5 ± 1.4 |
| **2008** | 34.7 | 0.0185 | 0.1759 | 0.00600 ± 0.0026 | 3.4 ± 1.4 |
| **2009** | 36.3 | 0.0193 | 0.1681 | 0.00636 ± 0.0023 | 3.8 ± 1.2 |
| **2010** | 34.4 | 0.0183 | 0.1755 | 0.00604 ± 0.0023 | 3.4 ± 1.3 |
| **2011** | 33.6 | 0.0179 | 0.1614 | 0.00589 ± 0.0025 | 3.6 ± 1.5 |
| **2012** | 34.6 | 0.0184 | 0.1550 | 0.00507 ± 0.0024 | 3.5 ± 1.5 |
| **1970*** | 69.2 | 0.0368 | 0.2370 | 0.0121 | 5.12 |

*estimated values given for 1970 (see main text for details)



**Figures**



**Figure 1** Geographical location of measurement stations used. Location point colours describe location type: red - urban airport; blue - urban; purple - rural/remote and green - rural airport. Also presented are mean wind rose statistics whole time period (approximately 60 years) for all eight stations.





**Figure 2** Historical trend of annual and seasonal visibility derived from daily (12 noon) observations by station: **a)** Aldergrove **b)** Heathrow, **c)** Leuchars, **d)** Nottingham, **e)** Plymouth, **f)** Ringway, **g)** Tiree, **h)** Waddington. Shading indicates changes in measurement methodology. For further details see the supplementary Table S1











**Figure 3** Mean monthly visibility and RH (Left-side) and average weekday visibility (Right-side) at all eight sites:

**a)** Aldergrove **b)** Heathrow **c)** Leuchars **d)** Nottingham **e)** Plymouth **f)** Ringway **g)** Tiree **h)** Waddington.













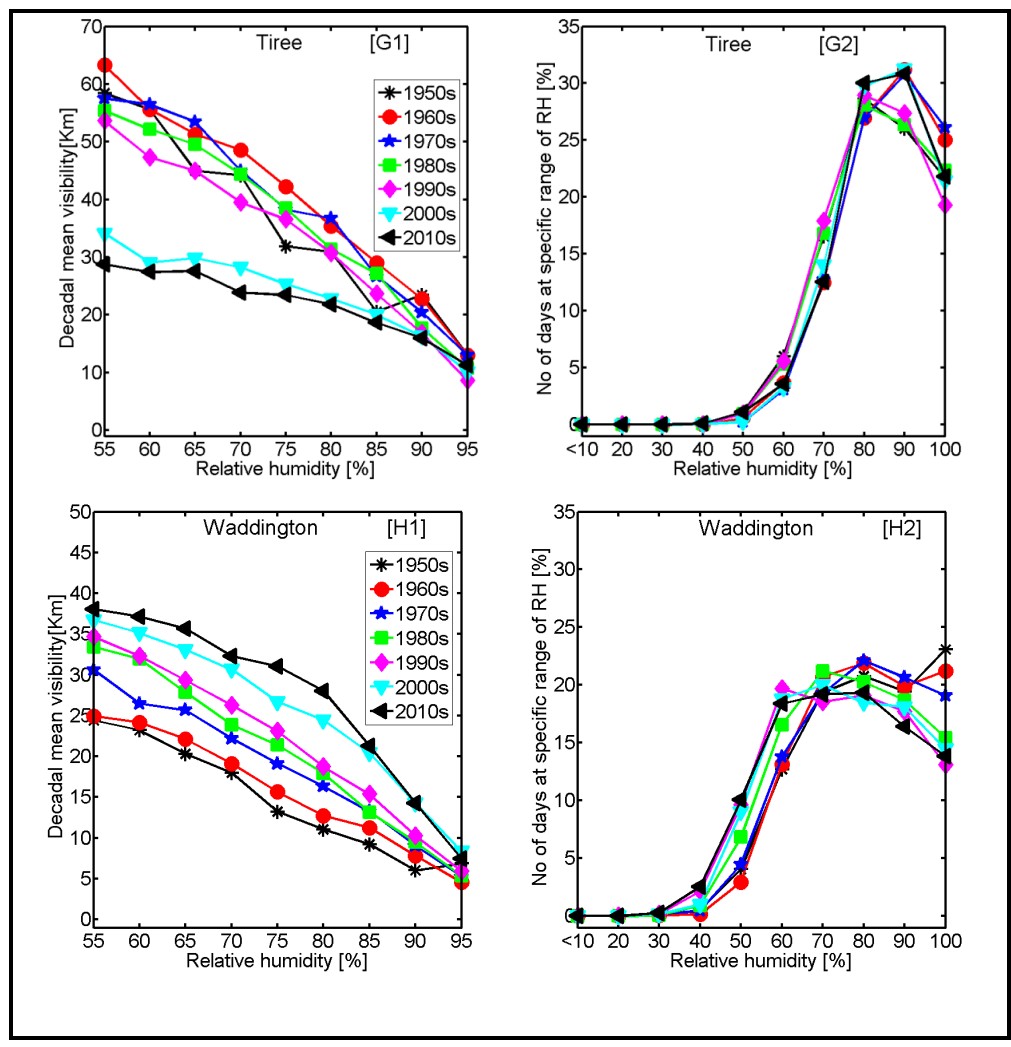

**Figure 4** Decadal visibility at specific range of relative humidity (left side) and number of days in % during different relative humidity (right side)





**Figure 5** Time-series of meteorological components relative humidity (RH), air temperature (T), wind speed ($w_s$), and wind direction ($w_d$) including visibility (V), where shaded lines show smooth fit line at 95 % confidence interval.



**Figure 6** Comparisons of modelled and observed visibility at specific range of RH using Eq. (4) at Heathrow station. The observed visibility is presented with standard error bars at 95 % confidence interval.





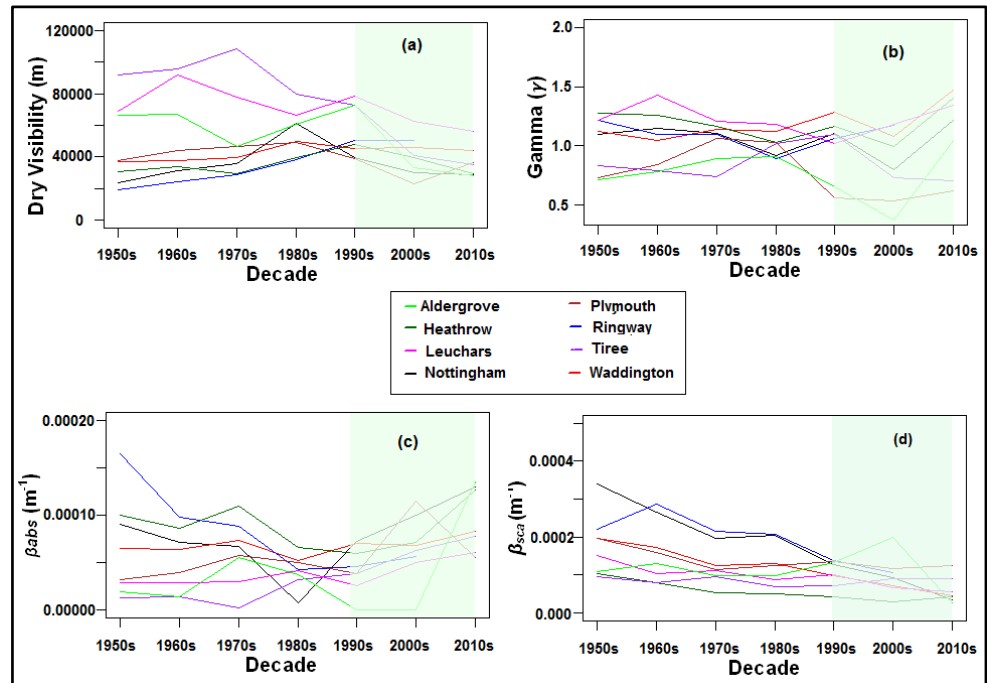

**Figure 7** Model output parameters **a)** Dry visibility, **b)** gamma and **c)** absorption coefficient and **d)** scattering coefficient at 75 % n.b. from 1950s to 2010s. The green shaded region shows the start of visiometer era at most of the stations (see supplementary Table S1 to see the starting year of visiometer measurement).

* See Supplementary Table S2 for model output parameter values including their uncertainties