# Peer review of "years of UK visibility measurements: impact of"

_Atmospheric Chemistry and Physics, 2016_

## Referee Comment (RC1) · Anonymous Referee #1 · 28 Sep 2016

This is an interesting study that measures 60 years of UK visibility in different environments (e.g. urban, rural, and marine) and shows the impact of meteorology and atmospheric pollutants on visibility. The authors use horizontal visibility data along with meteorological data from British Atmospheric Data Centre (BADC) to analyse UK visibility trends from 1950-2013. Although the authors extend the work of Doyle and Dorling (2002) to analyse UK visibility trends, but the reviewer find the dissimilarities of visibility values between Singh et al. (2016) and Doyle and Dorling (2002) results for the period of 1950 to 1997. The authors should explicitly describe why the visibility values presented in their study is different than the results from similar study by Doyle and Dorling (2002). In addition, specific descriptions on the explanations and discus-

sions on atmospheric sciences (e.g. reason for the reduction of air pollutants in urban areas) are insufficient. The author also develop a light extinction model for generating predictions of historic aerosol and gas scattering and absorbing properties. But the authors should provide more detailed discussions on the uncertainties which can arise in their modelling study. More specific comments are provided under 'Technical comments'. The manuscript is reasonably well written, but there are a lot of typographical errors throughout, which are noted under 'Editorial comments'. In my opinion, the manuscript is worth publishing, but some reviewer's concerns existed, which need to be addressed, then this should end up being a paper suitable for publication in ACP. Technical comments 1. Generally the reduction of visibility is found with increasing aerosol particles concentrations which has been explained briefly in introduction part of the paper. The results in this paper shows that the visibility of the urban areas has been improved year to year due to the reduction in air pollution for most of the monitoring stations in UK. But very little has been discussed about the possible reasons of the reduction in air pollution. Are they for cleaner fuel usage in the vehicles? Is it for increased deposition at the building surface due to the urbanisation and human population? Or do the authors have any other suggestion? In marine and rural environments, the natural emissions of aerosol precursors (e.g. DMS from ocean and terpenes from rural plants) are dominating which cannot be controlled. Do the authors think that this could be the reason for decreased visibility over time for marine and rural stations?

2. After comparing the visibility results of this paper with Doyle and Dorling (2002) results, the reviewer found the similar variation trend for the period of 1950-1995, but the visibility values are found to be lower for all stations in this study than Doyle and Dorling (2002) study. Why does this paper produce lower visibility values? No explanation/comparison has been shown in the paper.

3. 12 noon data has been taken as the daily data, however there could be the variation of the visibility throughout the day because of the variation of the meteorological parameters and the concentration of aerosol particles. These need to be discussed as

Interactive
comment

the uncertainties of their results. The authors excluded the data for 99% precipitation. How much percent data points of 99% precipitation? How the whole data analysis has been affected after excluding these data points? 4. Page 1: The reviewer think that some of the sentences (e.g. Moreover, degradation in visibility can be hazardous to . . . . . . . . . . . . ...sea and air accidents, The site locations include . . . . . . ..  marine environments, the model incorporates parameterizations . . . . . . . . . . ..and particle and gas adsorption) in Abstract are not meaningful. Instead, they can only be placed in introduction and methodology part. 5. Page 2 Line 3: The literature review (WMO, 2003) is very old. The authors should consider updating their literature review by using recent report by WMO (2015). There are some other places in the introduction part where the references can be updated. Reference: World Meteorological Organization, 2015: Manual on the Global Observing System. Volume I – Global aspects, (WMO-No. 544), Geneva.

6. Page 4, Line 32: The reader might be confused in many places of the manuscript as authors used 'human observation' and 'manual observation' for same meaning. As human observation is more common term for visibility measurement, the reviewer suggest the authors to change 'manual observation' to 'human observation' throughout the manuscript. 7. Page 5, Line 20-24: The authors claimed that at high visibility the automatic sensors perform sub-optimally at coastal site (e.g. Tiree) due to accumulation of sea salt residue. If this is the case, it will also be applicable for another coastal site, Leuchars. But the station Leuchars did not show any deviation when the measurement moved from manual to automation. How the authors will explain the different measurement behaviour for similar type of stations, Tiree and Leuchars? 8. Page 9, Line 4: What localized sources close to visiometer at the Plymouth site? Are they aerosol particles? 10. Page 10, line 9-13: The sentence is contradictory to the reviewer. The reviewer can see from the decadal polar plot that the visibility has been improved decade to decade when the wind comes from the south to east direction, but the reviewer doesn't understand how this is connected with the part "the higher wind speeds from the direction of Belfast leads to lower visibility over Aldregrove". Overall

the lower visibility in the south to east direction can suit with the above statement. 11. Page 12, Lines 10-15: Is there any specific reasons for higher visibilities on Friday for Leuchars and on Wednesday for Ringway? 12. Page 13, Line 5: It would strength the manuscript if the authors can show the relationship of the visibility with temperature. From Fig 5, the reviewer can't see any relationship between visibility and air temperature. 13. Figure 2: Measurement methodology Supplementary Table shows that the measurement was performed manually at Ringway from 2004 to onwards, but in the figure-2, there are no data points from 2004. And also the shading should be white as they are manual measurements. And for Heathrow, the red shading and blue shading are the measurements using the same instrument? 14. Figure 5: The reviewer doesn't think this figure is necessary, as most of them already shown in previous figures. Instead, this figure can be placed in the supplementary Information. However, the rose plot for annual average (for full data series) can be placed in the main manuscript which will be easier for reader to see the overall influence of wind speed and wind direction on visibility. The decadal seasonal polar plots can be kept in Supplementary Information. 15. Figure 7: The green shaded region has been shown from 1990s, but most of the stations start visiometer measurement from 2000. Will it be 2000s instead of 1990s?

Editorial Comments Page 1 Line 16: examples of urban areas are preferable. Page 2, Line 23: 'sharp changes' can be replaced by 'sharp decreases' Page 2, Line 25: 'describe' need to be replaced by 'described' Page 3, Line 23: 'decline' should be 'been declined' Page 4, Line 4: 'to' needs to be added in between 'help' and 'explain' Page 4, Line 5: 'They' should be replaced by 'We'. Page 4, Line 29: Is the term 'human observation' or 'human observer'? Page 5, Line 10, please delete 'than' Page 5, line 25, please add 'the' after assess

---

## Referee Comment (RC2) · Anonymous Referee #2 · 14 Oct 2016

General comments The study explores UK horizontal visibility, using observations from a number of stations of different characteristics. Actually the study extends the work by Doyle & Dorling (2002) who reported UK visibility improvement from 1950-1997 due to antipollution measures. The extension alone is not so useful as regards estimation of long term trends, since a very strong step change occurred after changes in observational methods. However, authors perform detailed analyses regarding meteorological influence, the role of RH, and develop a light extinction model which make the study interesting. Many points however need to be reconsidered, corrected and clarified.

Major comments The study updates the work by Doyle & Dorling 2002, who study UK visibility from 1950-1997. The same stations and the same visibility hour (12Z) have

been used in both studies. So, one would expect to see exactly the same values of visibility for the common period which is not true. In contrast, the authors estimate lower values almost in all stations. Is it because of different filters? Or averaging procedure? Following Sloane (1982), Doyle & Dorling exclude visibility values when RH >90%. The authors use another filter (99%) which means that they use more high RH days. I can suppose that this is the reason for the observed differences in the two studies. Some clarifications are required however. How was determined the filter 99%? The authors relate visibility with meteorology, however, precipitation is a fundamental parameter which is missing from this analysis. Precipitation increases RH, but also is related to scavenging of particles in the atmosphere, possibly improving visibility. Precipitation frequency than amount is more important indicator and consequent cleanup of the atmosphere is more important in these cases. So I am wondering if any relevant data are available from nearby stations. Averaging procedure of visibility is not mentioned. Which code/protocol has been used for human visibility observations? Since uncertainties are much higher in high visibility ranges (as you also mention in Page 5, line 5), visibility follows a rather geometric distribution. Did you use simple mean or a geometric mean for visibility? The method of measuring visibility changed from human observations to electronic visiometers. This was done at different times for each station. The impact of this change is dramatic as easily seen in Fig. 2. I would say that it is impossible under these circumstances to draw a conclusion for the long term trend of visibility. Are the two methods compared at any station? Is there any parallel period with human + electronic observations? This is a common procedure to evaluate and compare the two methods. If such parallel measurements are available, then authors need to make proper comparisons/calibration and provide a better transition from the first to the second period. The authors use wind roses from surface wind data and perform an extended analysis on visibility variation with respect to wind speed/direction. This is related to air mass origin and associated air pollution or RH sources. Although they perform a reasonable analysis, I think that additional information is required regarding local or long transport pollution from distant sources.
Frequently, surface winds reflect very local phenomena (breezes, circulations due to UHI effect, chanelling phenomena etc). Although authors refer to long range transport of air pollution from central Europe (for eastern sector) there is no information on long range transport (trajectories, frequencies etc). European emissions increased after the 1950s and decreased after the 1980s. Is UK unaffected from these changes? Is it all local pollution? A discussion on this is necessary. In general, information on local pollution sources and reasons for improving per sector is not adequate. Relative to this, in page 14, line 9, the authors seem to speculate.

How do you define good or poor visibility? In Fig 2 authors present long-term trends of the annual/seasonal visibility averages and find an overall positive trend in most stations. However, this cannot provide information on the relative improvement in different visibility ranges. Is the improvement higher in low, average or higher visibilities? I would like to see a frequency distribution of different visibility ranges for different sub-periods, which would be much more informative on visibility improvement.

In Fig. 5 the authors provide long term records of annual visibility and annual averages of different meteorological parameters. A comparison is attempted between variation of visibility and meteorological variables. I have some questions here. Annual visibility was calculated using daily measurements at 12Z. How other variables were averaged? Do averages refer to 24-hour periods? From the figure it comes out (visually) that visibility is anticorrelated (in low frequencies) with RH. However, RH changes do not refer to 12Z (I think) and also these changes are small enough (in the range of very few units of %, for instance from 75% to 78% or something like that). In the analysis of Fig. 4 such changes fall into the same RH category. What mean annual WD refers to? Is it prevailing wind direction? How was calculated? In the same figure, wind speed variability does not seem to be positively correlated (as expected) with visibility. Decreasing trends of wind speed in some stations are accompanied with increasing trends in visibility. Does it mean that wind speed is less influential? Perhaps a running correlation coefficient between visibility and other meteorological variables would be

**ACPD**
more informative on the influence of such variables and possible temporal changes of this influence. The relationship with air temperature is tentative. At urban area in particular, air temperature increase could refer to nocturnal increases due to urban heat island effect (but visibility refers to noon). Some clarifications are required.

Model: The authors present a model for light extinction, making a number of assumptions and simplifications. Which could be the cost (uncertainty arising from these assumptions)? Despite assumptions, the model has an absolutely perfect performance with observations. Any explanation? What about the other stations?

Page 4, line 5: The aim of the study is implemented? what do you means UK projections of meteorology (climate change? it is not clear). And what do you mean with pollution projections? Local or regional? What kind of projections? For which pollutants?

Minor comments Abstract, line 1: This is not always true, add meteorology factor. Abstract, Line 2. It can be removed from abstract Page 2, line 16: rearrange using chronological order In the analysis of week day variations of visibility , the information provided in Page 12, line 14 is confusing and I also think wrong (regarding the calculations). I do understand the meaning of this analysis. In Figures 3 (right side), it is better to use normalized values. For instance you can normalize values with the maximum visibility value for a direct estimation of % differences. Page 5, line 25: do you mean the sensor was not cleaned? How can you be sure that all other stations are cleaned properly?

**Technical comments**

Although English is in general good, some syntax errors exist in the paper. Missing comma in many cases make the text hard to understand. Figures quality needs to be improved. Use legends in Fig. 3 or use analogous (with variables) colors in the axis Fig3. Indicate in the legends what dashed lines represent (left side) and bars (right side).

**ACPD**

---

## Author Comment (AC1) · 18 Nov 2016

Response to reviewers for manuscript acp-2016-738: **60 years of UK visibility measurements: impact of meteorology and atmospheric pollutants on visibility**

We thank the reviewers for their time and excellent insights which have helped us to improve the manuscript. We now thank them in the acknowledgements.
We respond to all of the reviewers' points below. Responses are given in red.

**Anonymous Referee #1**

This is an interesting study that measures 60 years of UK visibility in different environments (e.g. urban, rural, and marine) and shows the impact of meteorology and atmospheric pollutants on visibility. The authors use horizontal visibility data along with meteorological data from British Atmospheric Data Centre (BADC) to analyse UK visibility trends from 1950-2013. Although the authors extend the work of Doyle and Dorling (2002) to analyse UK visibility trends, but the reviewer find the dissimilarities of visibility values between Singh et al. (2016) and Doyle and Dorling (2002) results for the period of 1950 to 1997. The authors should explicitly describe why the visibility values presented in their study is different than the results from similar study by Doyle and Dorling (2002). In addition, specific descriptions on the explanations and discussions on atmospheric sciences (e.g. reason for the reduction of air pollutants in urban areas) are insufficient. The author also develop a light extinction model for generating predictions of historic aerosol and gas scattering and absorbing properties. But the authors should provide more detailed discussions on the uncertainties which can arise in their modelling study. More specific comments are provided under 'Technical comments'. The manuscript is reasonably well written, but there are a lot of typographical errors throughout, which are noted under 'Editorial comments'. In my opinion, the manuscript is worth publishing, but some reviewer's concerns existed, which need to be addressed, then this should end up being a paper suitable for publication in ACP.

**Technical comments**

**1.** Generally the reduction of visibility is found with increasing aerosol particles concentrations which has been explained briefly in introduction part of the paper. The results in this paper shows that the visibility of the urban areas has been improved year to year due to the reduction in air pollution for most of the monitoring stations in UK. But very little has been discussed about the possible reasons of the reduction in air pollution. Are they for cleaner fuel usage in the vehicles? Is it for increased deposition at the building surface due to the urbanisation and human population? Or do the authors have any other suggestion? In marine and rural environments, the natural emissions of aerosol precursors (e.g. DMS from ocean and terpenes from rural plants) are dominating which cannot be

controlled. Do the authors think that this could be the reason for decreased visibility over time for marine and rural stations?

Thanks for this useful comment. The primary aim of this paper is not a detailed source of apportionment study; therefore we cannot be certain what caused the visibility changes at the different sites. Previous work in the literature has shown that changes in fuel use that came about after the clean Air Act are likely reasons. We now provide more information on the likely causes determining the visibility on Page 10 Line 6 to Page 10 Line 12 "Improved visibility at most of the sites is due to reduction in air pollution and the likely changes in fuel use and consumption that took place after 1956 Clean Air Act. The Clean Air Act was introduced with the aims of reducing smog, smoke and sulphur dioxide concentrations in the environment. In particular, the policy focused on industrial emission sources and its reduction (Williams, 2004). Recently, Harrison et al. (2015) shown that concentration of sulphur dioxide, coal smoke, nitrogen dioxide, suspended matter (black smoke) and PM were significantly reduced in the UK over last five decades as the result of switching to cleaner fuels after 1956 Clean Air Act."

**2.** After comparing the visibility results of this paper with Doyle and Dorling (2002) results, the reviewer found the similar variation trend for the period of 1950-1995, but the visibility values are found to be lower for all stations in this study than Doyle and Dorling (2002) study. Why does this paper produce lower visibility values? No explanation/comparison has been shown in the paper.

We agree that there is similar variation trend for the period of 1950-1997, with our work showing slightly lower visibility values compared to Doyle and Dorling (2002). Doyle and Dorling (2002) filtered data for 12 noon, relative humidity <90% and Present weather codes (PR code) of 00-05 in their statistical analysis for the period of 1950-1997. In our study we used mean averaging for statistical analysis, where we filtered data for 12 noon and relative humidity <99%. As discussed already in the manuscript PR codes were not available after 1997 and hence we could not use them (Page 4 Line 16 to Page 4 Line 21):

"Unfortunately the use of present weather codes largely ceased with the introduction of automated meteorological stations and insufficient PR codes were available after the year 1997. It is noted, that if the present weather codes were available they would have been useful to screen the data for rain or other precipitation events. Due to unavailability of present weather codes during required study period (1950-2012), data filtration was done on the bases of RH limits instead of PR codes. Data were removed when the relative humidity reading was > 99 % which is highly suggestive of rain or other precipitation events."

The differences in our filtering approach, compared to that used in Doyle and Dorling, is the reason why we have lower visibility values compared to Doyle and Dorling (2002). More explanation and comparison are now added in Page 9 Line16 to Page 9 Line 23 "A similar variation in visibility trends is observed for the period of 1950-1997, comparing with Doyle and Dorling (2002). However, this study reports overall lower visibility values when compared to Doyle and Dorling (2002). These differences are due to slightly different data filtering methodologies. Doyle and Dorling (2002) filtered data for 12 noon, relative humidity > 90% and PR codes of 00-05 in their statistical analysis for the period of 1950-1997. However, due to uncertainty and unavailability of PR code after 1997 we did not use these codes. Furthermore we performed mean averaging for statistical analysis, where data is filtered for 12 noon and relative humidity > 99 %. The details of uncertainty and unavailability of PR codes and used data filtration method are given in data and methodology sections".

**3.** 12 noon data has been taken as the daily data, however there could be the variation of the visibility throughout the day because of the variation of the meteorological parameters and the concentration of aerosol particles. These need to be discussed as uncertainties of their results. The authors excluded the data for 99% precipitation. How much percent data points of 99% precipitation? How the whole data analysis has been affected after excluding these data points?

Although analysis could be performed for any hour of the day, we chose 12 noon because as stated on Page 4 Line 22 manual observations of visibility can be affected by low light levels. Therefore, we chose a time when light levels were near their maximum.

As shown in Table R1 below the number of data points removed due to the filtering of data with RH >99% is very low. The filtered data accounts for 0.91 – 3.44 % of the total data dependent upon site location. Therefore removing these points does not make any significant difference. We now make the point in the manuscript on Page 4 Line 21 "Removal of data with RH > 99% removes between 0.91 – 3.44 % of the data dependent on site location".

Table R1

| Station | Total data points | Data points of above 90% precipitation | Data points of above 99% precipitation | % (data points of above 90% precipitation) | % (data points of above 99% precipitation) |
|---|---|---|---|---|---|
| Aldergrove | 23370 | 4491 | 619 | 19.25 % | 2.65 % |
| Heathrow | 23322 | 2460 | 292 | 10.53 % | 1.25 % |
| Leuchars | 20814 | 2839 | 190 | 13.63 % | 0.91 % |

| Nottingham | 20351 | 3749 | 620 | 18.42 % | 3.04 % |
|------------|-------|------|-----|---------|--------|
| Plymouth | 23183 | 4719 | 798 | 20.35 % | 3.44 % |
| Ringway | 20027 | 2140 | 195 | 10.68 % | 0.97 % |
| Tiree | 20412 | 4601 | 472 | 22.54 % | 2.31 % |
| Waddington | 23163 | 4014 | 710 | 17.33 % | 3.07 % |

**4.** Page 1: The reviewer think that some of the sentences (e.g. Moreover, degradation in visibility can be hazardous to . . .. . .. . .. . ...sea and air accidents, The site locations include . . .. . ... marine environments, the model incorporates parameterizations . . .. . .. . ..and particle and gas adsorption) in Abstract are not meaningful. Instead, they can only be placed in introduction and methodology part.

We think a brief description of visibility and the model strengthens the abstract by providing context for the paper.  No changes have been made.

**5.** Page 2 Line 3: The literature review (WMO, 2003) is very old. The authors should consider updating their literature review by using recent report by WMO (2015). There are some other places in the introduction part where the references can be updated. Reference: World Meteorological Organization, 2015: Manual on the Global Observing System. Volume I – Global aspects, (WMO-No. 544), Geneva.

Suggested changes have been implemented in the manuscript.

**6.** Page 4, Line 32: The reader might be confused in many places of the manuscript as authors used 'human observation' and 'manual observation' for same meaning. As human observation is more common term for visibility measurement, the reviewer suggest the authors to change 'manual observation' to 'human observation' throughout the manuscript.

Suggested changes have been implemented in the manuscript; where "manual" has changed to "human" throughout manuscript.

**7.** Page 5, Line 20-24: The authors claimed that at high visibility the automatic sensors perform sub-optimally at coastal site (e.g. Tiree) due to accumulation of sea salt residue. If this is the case, it will

also be applicable for another coastal site, Leuchars. But the station Leuchars did not show any deviation when the measurement moved from manual to automation. How the authors will explain the different measurement behaviour for similar type of stations, Tiree and Leuchars?

We agree that station Leuchars did not appear to show any deviation when the measurement moved from manual to automation. We suggest two possible reasons, firstly as shown in Figure 1, the average wind speed of Tiree is higher than Leuchars, which causes more sea salt to be generated and transported to the site. Secondly, as mentioned in the main manuscript Page 12 Line 28 that Tiree Island has a very flat landscape, which is not sheltered from the wind in any direction, thereby potentially allowing salt to accumulate more readily on the sensor.

**8.** Page 9, Line 4: What localized sources close to visiometer at the Plymouth site? Are they aerosol particles?

The following sentence has been added in Page 9 Line 10 about localized source at Plymouth site "(ship and traffic emissions from nearby ports and roads)".

**10.** Page 10, line 9-13: The sentence is contradictory to the reviewer. The reviewer can see from the decadal polar plot that the visibility has been improved decade to decade when the wind comes from the south to east direction, but the reviewer doesn't understand how this is connected with the part "the higher wind speeds from the direction of Belfast leads to lower visibility over Aldregrove". Overall the lower visibility in the south to east direction can suit with the above statement.

We have revised the sentence, which now reads "It is clearly seen that visibility has improved the most when wind comes from the south to east direction which covers mainland urban areas such as Belfast, the major regional city." (Page 10 Line 30)

**11.** Page 12, Lines 10-15: Is there any specific reasons for higher visibilities on Friday for Leuchars and on Wednesday for Ringway?

We checked the data on those two days to investigate any extreme high visibility values, but found no specific resasons. We note that the visibility changes on these days are slight compared to the weekend effect. The main focus of this graph is to show the difference between weekday and weekend visibility. This graph has now been replaced with normalized weekday visibility graph in

response to one of Reviewer 2's questions, where mean weekday visibility normalized to Sunday mean values provides a direct estimate of the percentage differences in weekday visibility values.

**12.** Page 13, Line 5: It would strength the manuscript if the authors can show the relationship of the visibility with temperature. From Fig 5, the reviewer can't see any relationship between visibility and air temperature.

Most sites show clear anti-correlation between temperature and relative humidity as is expected under UK meteorological conditions. Hence a correlation also exists between temperature and visibility. We have now added figure S5 for correlation statistics. The following sentence has been added in Page 13 Line 29 "The correlation statistics between visibility, relative humidity, air temperature and wind speed are provided for all stations in supplementary Fig. S5."

**13.** Figure 2: Measurement methodology Supplementary Table shows that the measurement was performed manually at Ringway from 2004 to onwards, but in the figure-2, there are no data points from 2004. And also the shading should be white as they are manual measurements. And for Heathrow, the red shading and blue shading are the measurements using the same instrument?

We agree, we have mentioned in Supplementary Table S1 that visibility measurement was performed manually at Ringway from 2004 onward; however, very limited numbers of observations at 12 noon are available for statistical analysis and hence we have not able to show annual and seasonal variability after 2004.

There was a mistake in the shading; it has now been replaced. For Heathrow red shading and blue shading are the measurements using the different instrument. More details are now added in Figure 2 caption; Page 24 "Shading indicates changes in measurement methodology, where white is human observation, while blue and red are automated observation using different instruments."

**14.** Figure 5: The reviewer doesn't think this figure is necessary, as most of them already shown in previous figures. Instead, this figure can be placed in the supplementary Information. However, the rose plot for annual average (for full data series) can be placed in the main manuscript which will be easier for reader to see the overall influence of wind speed and wind direction on visibility. The decadal seasonal polar plots can be kept in Supplementary Information.

The suggested change has been implemented, where Figure 5 has been placed in the supplementary Information as Figure S4. Figure 1 shows the wind rose plot for annual average (for full data series) and we think should remain in the main manuscript.

**15.** Figure 7: The green shaded region has been shown from 1990s, but most of the stations start visiometer measurement from 2000. Will it be 2000s instead of 1990s?

The suggested change has been implemented in Figure 7, where green shade has been changed from 1990s to 2000s.

**Editorial Comments**

Page 1 Line 16: examples of urban areas are preferable.

Suggested change has been implemented in the manuscript

Page 2, Line 23: 'sharp changes' can be replaced by 'sharp decreases'

Suggested change has been implemented in the manuscript

Page 2, Line 25: 'describe' need to be replaced by 'described'

Suggested change has been implemented in the manuscript

Page 3, Line 23: 'decline' should be 'been declined'

Suggested change has been implemented in the manuscript

Page 4, Line 4: 'to' needs to be added in between 'help' and 'explain'

Suggested change has been implemented in the manuscript

Page 4, Line 5: 'They' should be replaced by 'We'.

Thanks for pointing out this. This sentence was unclear. We now state the following "A new model is also presented which can aid in future visibility prediction under different climate and pollution scenarios."

Page 4, Line 29: Is the term 'human observation' or 'human observer'?

Suggested change has been implemented, where human observation has changed with human observer

Page 5, Line 10, please delete 'than'

Suggested change has been implemented in the manuscript

Page 5, line 25, please add 'the' after assess

Suggested changes have been implemented in the manuscript

**Anonymous Referee #2**

General comments The study explores UK horizontal visibility, using observations from a number of stations of different characteristics. Actually the study extends the work by Doyle & Dorling (2002) who reported UK visibility improvement from 1950-1997 due to antipollution measures. The extension alone is not so useful as regards estimation of long term trends, since a very strong step change occurred after changes in observational methods. However, authors perform detailed analyses regarding meteorological influence, the role of RH, and develop a light extinction model which make the study interesting. Many points however need to be reconsidered, corrected and clarified.

**Major Comments**

The study updates the work by Doyle & Dorling 2002, who study UK visibility from 1950-1997. The same stations and the same visibility hour (12Z) have been used in both studies. So, one would expect to see exactly the same values of visibility for the common period which is not true. In contrast, the authors estimate lower values almost in all stations. Is it because of different filters? Or averaging procedure? Following Sloane (1982), Doyle & Dorling exclude visibility values when RH >90%. The authors use another filter (99%) which means that they use more high RH days. I can suppose that this is the reason for the observed differences in the two studies. Some clarifications are required however. How was determined the filter 99%?

We agree that there is similar variation trend for the period of 1950-1997, with our work showing slightly lower visibility values compared to Doyle and Dorling (2002). Doyle and Dorling (2002) filtered data for 12 noon, relative humidity <90% and Present weather codes (PR code) of 00-05 in their statistical analysis for the period of 1950-1997. In our study we used mean averaging for statistical analysis, where we filtered data for 12 noon and relative humidity <99%. As we mentioned in manuscript PR codes were not available after 1997 and hence we could not use them (Page 4 Line 16 to Page 4 Line 21):

"Unfortunately the use of present weather codes largely ceased with the introduction of automated meteorological stations and insufficient PR codes were available after the year 1997. It is noted, that if the present weather codes were available they would have been useful to screen the data for rain or other precipitation events. Due to unavailability of present weather codes during required study period (1950-2012), data filtration was done on the bases of RH limits instead of PR codes. Data were removed when the relative humidity reading was > 99 % which is highly suggestive of rain or other precipitation events."

The differences in our filtering, compared to that used in Doyle and Dorling, is the reason why we have little lower visibility values from Doyle and Dorling (2002). More explanation and comparison are now added in Page 9 Line16 to Page 9 Line 23 "A similar variation in visibility trends is observed for the period of 1950-1997, comparing with Doyle and Dorling (2002). However, this study reports overall lower visibility values when compared to Doyle and Dorling (2002). These differences are due to slightly different data filtering methodologies. Doyle and Dorling (2002) filtered data for 12 noon, relative humidity > 90% and PR codes of 00-05 in their statistical analysis for the period of 1950-1997. However, due to uncertainty and unavailability of PR code after 1997 we did not use these codes. Furthermore we performed mean averaging for statistical analysis, where data is filtered for 12 noon and relative humidity > 99 %. The details of uncertainty and unavailability of PR codes and used data filtration method are given in data and methodology sections".

The choice of RH filter (RH<99%) was chosen because this paper was interested in deriving aerosol parameters from the visibility data. In particular, to be able to understand changes in particle hygroscopicity, via the gamma ($\gamma$) parameter, the analysis needs a wide range of RH to fit the model as successfully as possible. If we exclude visibility values when RH >90%, we are losing likely 10-22 % data points (please see Table R1 in Reviewer 1 replied comments). This is the reason we excluded visibility values when RH >99%.

The authors relate visibility with meteorology, however, precipitation is a fundamental parameter which is missing from this analysis. Precipitation increases RH, but also is related to scavenging of

particles in the atmosphere, possibly improving visibility. Precipitation frequency than amount is more important indicator and consequent cleanup of the atmosphere is more important in these cases. So I am wondering if any relevant data are available from nearby stations.

We agree with the referee it would be nice to investigate the role of precipitation upon visibility. Unfortunately in all cases collocated precipitation data are not available for the required study period; therefore, precipitation could not be a focus of this manuscript.

Averaging procedure of visibility is not mentioned. Which code/protocol has been used for human visibility observations? Since uncertainties are much higher in high visibility ranges (as you also mention in Page 5, line 5), visibility follows a rather geometric distribution. Did you use simple mean or a geometric mean for visibility?

The method of measuring visibility changed from human observations to electronic visiometers. This was done at different times for each station. The impact of this change is dramatic as easily seen in Fig. 2. I would say that it is impossible under these circumstances to draw a conclusion for the long term trend of visibility. Are the two methods compared at any station? Is there any parallel period with human + electronic observations? This is a common procedure to evaluate and compare the two methods. If such parallel measurements are available, then authors need to make proper comparisons/calibration and provide a better transition from the first to the second period.

Simple mean method is used for the visibility averaging in the analysis, now has mentioned in Page 5 Line 36. We now also produce boxplots showing the median, interquartile range, outliers etc… in the supplementary material. We note that the median is often close in value to the mean average and the trends remain the same. The following sentences have been added in Page 10 Line 3 to Line 5 "The improvement in median visibility at most of the sites can be seen in supplementary Fig. S1. Boxplots of the decadal visibility are also produced showing the median, interquartile range, outliers etc. (see supplementary Figure S2)."

The details of visibility observations are provided within the UK Met Office guidelines (https://badc.nerc.ac.uk/data/ukmo-midas/ukmo_guide.html).

"Visibility is defined as the greatest distance at which an object in daylight can be seen and recognised, or at night could be seen and recognised if the illumination was raised to daylight levels. Observations should be made at ground level not from observation towers or roof tops. The long standing method of observation has been estimation by the observer using known fixed reference points, such as trees or buildings, which stand out well against the background. Each reference point

should subtend an angle of at least $0.5^o$ at the eye. Estimation of visibility at night is prone to greatest error and should, ideally, by performed with the aid of suitable fixed lights. Visibility estimates on airfields, where accuracy is of particular importance, are often aided in this way. On occasions when the visibility varies in different directions, the minimum value should be reported in the main part of the message and this is the value stored in MIDAS. The guidance to observers at coastal stations states that only visibility over land should be reported; any differing values over the sea being noted in the remarks column of the weather register though it is not clear how closely this practice is followed at voluntary stations."

For more detail in manuscript, the following line has been added in the manuscript Page 4 Line 29 to Page 4 Line 31 as "More details of visibility observations method are found in UK Met Office Surface Data Users guidelines (https://badc.nerc.ac.uk/data/ukmo-midas/ukmo_guide.html)."

Unfortunately, there are no periods when both human and electronic observations were taken in parallel, therefore we cannot perform a comparison between the two measurement types. To support the conclusion of long term trend in visibility, we have included an additional Figures S1 and S2 in the supplementary material, which show density distribution and decadal box plots of visibility at each station.

The authors use wind roses from surface wind data and perform an extended analysis on visibility variation with respect to wind speed/direction. This is related to air mass origin and associated air pollution or RH sources. Although they perform a reasonable analysis, I think that additional information is required regarding local or long transport pollution from distant sources. Frequently, surface winds reflect very local phenomena (breezes, circulations due to UHI effect, chanelling phenomena etc). Although authors refer to long range transport of air pollution from central Europe (for eastern sector) there is no information on long range transport (trajectories, frequencies etc). European emissions increased after the 1950s and decreased after the 1980s. Is UK unaffected from these changes? Is it all local pollution? A discussion on this is necessary. In general, information on local pollution sources and reasons for improving per sector is not adequate. Relative to this, in page 14, line 9, the authors seem to speculate.

Similar to our response to reviewer 1, we note that this paper does not attempt to give a detailed source apportionment of pollution in the UK. An analysis of very local phenomena and long range transport trajectories and frequencies over a multi decade time period represents another major piece of work and is beyond the scope of this study.

Whilst we have not performed trajectory analysis, we have noted previous studies have shown synoptic scale pollution events from Europe affecting UK pollution levels. This information is given in Page 12 Line 19 to 23. We have also added a more recent reference in this section (Crilley et al., 2015) which provides a similar conclusion.

How do you define good or poor visibility? In Fig 2 authors present long-term trends of the annual/seasonal visibility averages and find an overall positive trend in most stations. However, this cannot provide information on the relative improvement in different visibility ranges. Is the improvement higher in low, average or higher visibilities? I would like to see a frequency distribution of different visibility ranges for different sub-periods, which would be much more informative on visibility improvement.

We have now included density distribution graph of decadal visibility for each site in the supplement (Figure S1), where we can clearly see the improvement in median visibility. We have also included boxplot in the supplement to show the decadal changes in visibility over last six decades (Figure S2). The following sentences have been added in Page 10 Line 3 to Line 5 "The improvement in median visibility at most of the sites can be seen in supplementary Fig. S1. Boxplots of the decadal visibility are also produced showing the median, interquartile range, outliers etc. (see supplementary Figure S2)." We have added a definition of poor visibility "(< 2.0 km, (Founda et al., 2016))" in Page 2 Line 5.

In Fig. 5 the authors provide long term records of annual visibility and annual averages of different meteorological parameters. A comparison is attempted between variation of visibility and meteorological variables. I have some questions here. Annual visibility was calculated using daily measurements at 12Z. How other variables were averaged? Do averages refer to 24-hour periods? From the figure it comes out (visually) that visibility is anticorrelated (in low frequencies) with RH. However, RH changes do not refer to 12Z (I think) and also these changes are small enough (in the range of very few units of %, for instance from 75% to 78% or something like that). What mean annual WD refers to? Is it prevailing wind direction? How was calculated? In the same figure, wind speed variability does not seem to be positively correlated (as expected) with visibility. Decreasing trends of wind speed in some stations are accompanied with increasing trends in visibility. Does it mean that wind speed is less influential? Perhaps a running correlation coefficient between visibility and other meteorological variables would be more informative on the influence of such variables and possible temporal changes of this influence. The relationship with air temperature is tentative. At

urban area in particular, air temperature increase could refer to nocturnal increases due to urban heat island effect (but visibility refers to noon). Some clarifications are required.

Yes, annual visibility was measured at 12 noon. The other variables such as RH, air temperature, wind speed and wind direction are also average values for 12 noon. This information has now been included in supplementary Figure S4 caption, which has been removed from the main manuscript (Figure 5) to supplement as per suggestion of Reviewer 1.

Wd refers to the prevailing wind direction. More explanation is now added in Page 6 Line18 "These calculations were performed using the timePlot function in the openair package for R statistical program, which works on vector functions for wind direction averaging."

We looked at the correlation between different meteorological variables. This data is now provided in Supplementary Figure S5. Statistically significant correlations are observed between visibility and air temperature, relative humidity and wind speed.

Model: The authors present a model for light extinction, making a number of assumptions and simplifications. Which could be the cost (uncertainty arising from these assumptions)? Despite assumptions, the model has an absolutely perfect performance with observations. Any explanation? What about the other stations?

We understand the excellent agreement between model and observations as evidence that the model is capturing the key features of the underlying physics of atmospheric visibility. We were very pleased by this result! Whilst the model is good – it is not perfect – and there are a number of mismatches between model and observation albeit slight. We state clearly in the manuscript the assumptions we make, in particular, we state the following on Page 8 Line 1 "To reduce the number of parameters within Eq. (5), it is assumed that $\beta_{abs}(RH) = \beta_{abs}(dry)$. This assumption always holds for gas absorption; and it is largely true for aerosol particles as well, although it is noted that particle absorption can increase due to lensing effects in mixed phase aerosol, and this lensing effect will be affected by aerosol water content e.g. (Lack and Cappa, 2010)."

Page 4, line 5: The aim of the study is implemented? what do you means UK projections of meteorology (climate change? it is not clear). And what do you mean with pollution projections? Local or regional? What kind of projections? For which pollutants?

This sentence was unclear. We now state the following "A new model is also presented which can aid in future visibility prediction under different climate and pollution scenarios." We hope the model parameterization will be used by many future researchers who will find varied uses for it. For example, we can imagine that effect of both local and regional pollution upon visibility could be investigated. The effect of future RH predictions under climate change conditions could also be investigated.

**Minor Comments**

Abstract, Line 2. It can be removed from abstract

We believe this sentence is useful to provide context to the study and have not removed it.

Page 2, line 16: rearrange using chronological order

Thanks for this comment. We have found that ACP referencing format does not support chronological order. We have used ACP templates in EndNote, hence could not arrange the references in chronological order.

In the analysis of week day variations of visibility , the information provided in Page 12, line 14 is confusing and I also think wrong (regarding the calculations). I do understand the meaning of this analysis.

The calculation was wrong. We have simplified the text to the following - "Lower traffic and industrial emissions at the weekend are the likely reasons for better visibility at the weekend due to less pollutant emissions. The inherent assumption in this analysis is that traffic is higher during week days compared to the weekend. It is noted that visibility tends to peak on Sunday (rather than both Saturday and Sunday) and this may reflect the non-negligible timescale required for pollutant removal by wind driven dispersion, i.e. the build-up of pollution during weekdays is not fully dispersed until Sunday. The same argument explains why visibility is typically higher on Mondays compared to the other weekdays later in the week."

In Figures 3 (right side), it is better to use normalized values. For instance you can normalize values with the maximum visibility value for a direct estimation of % differences.

Figure 3 has been updated to use normalized values for the day of the week graph, where mean weekday visibility is normalized to the mean Sunday values for direct estimate of the percentage differences in weekday visibility values.

Page 5, line 25: do you mean the sensor was not cleaned? How can you be sure that all other stations are cleaned properly?

Unfortunately, we have no record for when they were cleaned, however we were informed by the UK Met Office that protocols were followed with regards to cleaning after high aerosol sea salt loads at coastal stations. As we described in the main manuscript, Tiree Island has a very flat landscape, which is not sheltered from the wind in any direction, possibly allowing salt to accumulate.

**Technical Comments**

Although English is in general good, some syntax errors exist in the paper. Missing comma in many cases make the text hard to understand. Figures quality needs to be improved. Use legends in Fig. 3 or use analogous (with variables) colors in the axis Fig3. Indicate in the legends what dashed lines represent (left side) and bars (right side).

Thanks for highlighting this. Suggested changed has been implemented in manuscript, where the typos in the manuscript have now been remedied. The higher resolution figures will be uploaded with the final manuscript. As per your suggestion we have also included analogous colours for variables in the Figure 3.

**References**

Crilley, L. R., Bloss, W. J., Yin, J., Beddows, D. C., Harrison, R. M., Allan, J. D., Young, D. E., Flynn, M., Williams, P., and Zotter, P.: Sources and contributions of wood smoke during winter in London: assessing local and regional influences, Atmospheric Chemistry and Physics, 15, 3149-3171, 2015.
Doyle, M., and Dorling, S.: Visibility trends in the UK 1950–1997, Atmospheric Environment, 36, 3161-3172, 2002.
Founda, D., Kazadzis, S., Mihalopoulos, N., Gerasopoulos, E., and Lianou, M.: Long-term visibility variation in Athens (1931–2013): A proxy for local and regional atmospheric aerosol loads, Atmos. Chem. Phys. Discuss., doi:10.5194/acp-2015-1025, in review, 2016., 2016.
Harrison, R. M., Pope, F. D., and Shi, Z.: Trends in Local Air Quality 1970–2014, Still Only One Earth: Progress in the 40 Years Since the First UN Conference on the Environment, 2015, 58,
Lack, D., and Cappa, C.: Impact of brown and clear carbon on light absorption enhancement, single scatter albedo and absorption wavelength dependence of black carbon, Atmospheric Chemistry and Physics, 10, 4207-4220, 2010.

Williams, M.: Air pollution and policy—1952–2002, Science of the total environment, 334, 15-20, 2004.

---

## Author Comment (AC2) · 18 Nov 2016

Please see attachment.

Please also note the supplement to this comment:
http://www.atmos-chem-phys-discuss.net/acp-2016-738/acp-2016-738-AC2-supplement.pdf

---

## Author Response (AR2)

Response to reviewers for manuscript acp-2016-738: **60 years of UK visibility measurements: impact of meteorology and atmospheric pollutants on visibility**

We thank the reviewer and editor for their time and excellent insights which have helped us to again improve the manuscript.

We respond to all of the reviewers' points below. Responses are given in blue.

The authors provided a new version of the manuscript, trying to correspond to both reviewer's comments and suggestions. In this revised version the authors improved their analysis mainly by providing additional missing information (actually certain clarifications)- as suggested by the reviewers - and also provided additional graphs in the supplementary materials.

I am a little disappointed by the fact that the authors did not try to find precipitation data (mainly precipitation frequency) in the areas of interest and correlate it with clean up of the atmospheric pollutants and possible visibility improvement. There are so many studies) on historical precipitation data in the UK (e.g. Alexander, L.V. and Jones, P.D. (2001) Updated precipitation series for the U.K. and discussion of recent extremes, Atmospheric Science Letters doi:10.1006/asle.2001.0025)…..and I think that there must be a very dense network of precipitation data in the UK.

Further investigation yielded 12 noon rainfall data for all sites for the majority of the study period. We now integrate this data and the conclusions resulting from it into the manuscript. The rainfall data set is acquired from the same repository as the other meteorological data and as is detailed as such in the data section of the paper. We also provide two new figures for the supplementary material.

In the manuscript we now make the following statement on Page 10 Line 13 to 19 "Rainfall data has been used to investigate the impact on visibility for all 8 study stations. Daily rainfall data from 12 noon averaged over each year is shown in supplementary Figure S3. Supplementary Figure S4 shows a comparison between annual average visibility that has been filtered for when rainfall is present (hourly rainfall > 0 mm) and non-filtered data. The percentage of data removed by filtering for rain accounts for 8 - 13% of the total data dependent upon the site location, with the Tiree and Aldergrove sites having the highest percentage of rainfall. It is observed that filtering for rainfall only results in very small visibility increases for some stations. Overall the effect is negligible in most circumstances. Therefore the non-filtered data is used in this study. "

'Figure S5' in supplement is not a 'Figure' but a 'Table' and should be referred accordingly. Suggested change has been implemented

Regarding averaging procedures, visibility protocols etc., the authors have added the following in their response: 'The details of visibility observations are provided within the UK Met Office guidelines(https://badc.nerc.ac.uk/data/ukmo-midas/ukmo_guide.html).'Looking in this site, I have seen the following : '….Visibility is reported in m or km and is stored in MIDAS in dam. In the SYNOP message a non-linear code is used giving a reporting precision of 30m (30m to 100 m); 100m (100 m to 5 km); 1km (5 km to 30 km) and 5 km (30 km to 70 km). There is a further coarser reporting code for use where there are few visual reference points which is principally used at sea. The accuracy requirement for observations of visibility from the synoptic network is +10%. Where visibility is measured at climatological stations the accuracy achieved is generally less that this value….'So I can understand (as I expected) that there is a visibility scale resulting to much higher uncertainty in higher visibility rates. For this reason I had already asked about 'averaging' procedures. "How do you define good or poor visibility? In Fig 2 authors present long-term trends of the annual/seasonal visibility averages and find an overall positive trend in most stations. However, this cannot provide information on the relative improvement in different visibility ranges. Is the improvement higher in low, average or higher visibilities? I would like to see a frequency distribution of different visibility ranges for different sub-periods, which would be much more informative on visibility improvement." To answer the above comment, the authors produced pdfs (probability density functions) for each station and each decade (Figure S1 of supplement material). However, although the (long) figure look detailed, it is not so convenient for comparison purposes. Moreover, in this figure the authors do not examine' ranges' of visibility but just visibility values. I think that the (too long) figure should be replaced by a more simple and 'easy to read' figure. For, instance I suggest a few visibility ranges (e.g. < 1km, 1-5, 5-30, 30-70, > 70) as indicated above) and fewer sub-periods, so you can produce histograms with frequencies (%) for each 'visibility range' and for each sub-period in the same graph for each stations. This will keep the number of figures at minimum and also enable comparison between different visibility ranges and different subperiods in the same plot for each station. This wil also show if improvement of visibility ia more important in ;high visibility' levels' in ;low visibility levels, or in all ranges.

We have now changed supplementary Figure S1 to include all decades, from a single location, onto a single plot to provide a figure that is easier to read. The plots show that the visibility improvement predominantly manifests itself in medium visibility range.  We produced frequency (histograms) plots

as suggested for different visibility ranges, which can be seen below (Figure R1), however we feel that the density plots provided in the supplementary material are clearer and we would prefer to use these. In our view, multiple histograms on the same plot are messy and difficult to interpret.

[Figure]

**Figure R1** Frequency of different visibility ranges for different sub periods (1950s to 2010s) for 8 sites.

[revised manuscript text omitted]

**Heathrow**

[Figure]

[Figure]

[Figure]

Nottingham

[Figure]

**Plymouth**

Mean visibility (km)

**Ringway**

[Figure]

**Mean visibility (km)**

**Tiree**

[Figure]

[Figure]

**Figure S7** Decadal seasonal polar plots for all eight stations for 1950s, 1960s, 1970s, 1980s, 1990s, 2000s and 2010s (left to right). * represents years where visiometer measured data is included.

[Figure]

**Figure S8 (a)** Scattering coefficient (*βsca*), **(b)** total extinction coefficient (*βext*) and **(c)** contribution of scattering coefficient in total extinction coefficient at Heathrow. Estimates of error are not included here to improve clarity.

[Figure]

**Figure S9** Model output parameters **a)** absorption coefficient ($\beta_{abs}$), **b)** Gamma ($\gamma$), and **c)** dry visibility at different seasons for Heathrow site.

[Figure]

**Figure S10** Decadal observed visibility at 70 % RH (range 67.5 -72.5 %) for Heathrow site. Error bars represent standard error at 95 % confidence interval.

**Supplementary Tables**

**Table S1** Method of visibility measurement at different station with its used time period, where present indicates the sensor is still installed and being used.

| Station Name | Method/ Sensor/ Equipment Type Name with their working period | | | | | |
|---|---|---|---|---|---|---|
| | **Manually** | | **Sensor 1** | | **Sensor 2** | |
| **Aldergrove** | | | **VISMETER -BELFORT 6230A** | | **PRESENT WEATHER SENSOR - FD12P** | |
| | 01/01/1926 | 24/01/2003 | 24/01/2003 | 28/08/2012 | 28/08/2012 | Present |
| **Heathrow** | | | **VISMETER -BELFORT 6230A** | | **VISMETER -BELFORT (Replaced with new one)** | |
| | 01/01/1947 | 01/01/2000 | 01/01/2000 | 15/06/2005 | 15/06/2005 | Present |
| **Ringway** | | | **VISIBILITY: VISIOMETER** | | **Manually** | |
| | 01/01/1941 | 01/01/2000 | 01/01/2000 | 01/11/2004 | 01/11/2004 | Present |
| **Nottingham** | | | **VISMETER -BELFORT 6230A** | | ------------------------------------- | |
| | 01/01/1941 | 01/01/2000 | 01/01/2000 | Present | | |
| **Plymouth** | | | **VISMETER -BELFORT 6230A** | | **PRESENT WEATHER SENSOR - FD12P** | |
| | 01/01/1920 | 23/01/1997 | 23/01/1997 | 16/12/2010 | 16/12/2010 | Present |
| **Tiree** | | | **VISMETER -BELFORT 6230A** | | ------------------------------------- | |
| | 01/01/1926 | 16/12/2010 | 16/12/2010 | present | | |
| **Leuchars** | | | **VISMETER -BELFORT 6230A** | | ------------------------------------- | |
| | 01/01/1921 | 16/12/2010 | 16/12/2010 | present | | |
| **Waddintgon** | | | **VISMETER -BELFORT 6230A** | | ------------------------------------- | |
| | 01/01/1946 | 01/01/2000 | 01/01/2000 | present | | |

**Table S2** Correlation coefficient (r) values between different variables, where daily data at 12 noon was used for calculation for last six decades

**Aldergrove**

|            | Visibility | RH      | Temp    | Wind speed |
|------------|------------|---------|---------|------------|
| Visibility | 1          |         |         |            |
| RH         | -.519**    | 1       |         |            |
| Temp       | .199**     | -.373** | 1       |            |
| Wind speed | .095**     | -.028** | -.050** | 1          |

**Heathrow**

|            | Visibility | RH      | Temp   | Wind speed |
|------------|------------|---------|--------|------------|
| Visibility | 1          |         |        |            |
| RH         | -.542**    | 1       |        |            |
| Temp       | .322**     | -.540** | 1      |            |
| Wind speed | .261**     | -.084** | -0.008 | 1          |

**Leuchars**

|            | Visibility | RH      | Temp    | Wind speed |
|------------|------------|---------|---------|------------|
| Visibility | 1          |         |         |            |
| RH         | -.688**    | 1       |         |            |
| Temp       | .179**     | -.353** | 1       |            |
| Wind speed | .124**     | -.208** | -.084** | 1          |

**Norringham**

|            | Visibility | RH      | Temp    | Wind speed |
|------------|------------|---------|---------|------------|
| Visibility | 1          |         |         |            |
| RH         | -.583**    | 1       |         |            |
| Temp       | .299**     | -.511** | 1       |            |
| Wind speed | .272**     | -.072** | -.054** | 1          |

**Plymouth**

|            | Visibility | RH      | Temp    | Wind speed |
|------------|------------|---------|---------|------------|
| Visibility | 1          |         |         |            |
| RH         | -.589**    | 1       |         |            |
| Temp       | .185**     | -.186** | 1       |            |
| Wind speed | .220**     | -.059** | -.073** | 1          |

**Ringway**

|            | Visibility | RH      | Temp    | Wind speed |
|------------|------------|---------|---------|------------|
| Visibility | 1          |         |         |            |
| RH         | -.549**    | 1       |         |            |
| Temp       | .342**     | -.423** | 1       |            |
| Wind speed | .269**     | -.070** | -.018** | 1          |

**Tiree**

|            | Visibility | RH      | Temp    | Wind speed |
|------------|------------|---------|---------|------------|
| Visibility | 1          |         |         |            |
| RH         | -.612**    | 1       |         |            |
| Temp       | .041**     | -.389** | 1       |            |
| Wind speed | .331**     | -.099** | -.226** | 1          |

**Waddington**

|            | Visibility | RH      | Temp   | Wind speed |
|------------|------------|---------|--------|------------|
| Visibility | 1          |         |        |            |
| RH         | -.633**    | 1       |        |            |
| Temp       | .340**     | -.550** | 1      |            |
| Wind speed | .232**     | -.091** | -.015* | 1          |

\* **Statistically significant value (p < 0.05)**

\*\* **Statistically significant value (p < 0.01)**

**Temp- Air Temperature          RH- Relative Humidity**

**Table S3** Model output parameters (*Vis(dry), βabs,* Gamma (*γ*) and *βsca)*

[revised manuscript text omitted]